

# OpenMindat v1.0.0 R package: A machine interface to Mindat open data to facilitate data-intensive geoscience discoveries

Xiang Que[1], Jiyin Zhang[1], Weilin Chen[1], Jolyon Ralph [2], Xiaogang Ma[1]

[1]Department of Computer Science, University of Idaho, Moscow, ID 83844, USA
[2]Hudson Institute of Mineralogy, Keswick, VA 22947, USA

*Correspondence to*: Xiaogang Ma (max@uidaho.edu)

**Abstract:** Powered by data-driven knowledge discovery technologies such as machine learning and deep learning, increasingly exciting patterns are discovered in complex earth science big data. One of the world's most enormous treasure troves of mineral databases, Mindat ("mindat.org"), contains vast amounts of knowledge that are yet to be mined. Through a project called

OpenMindat, an application programming interface (API) to enable open data query and access from Mindat had been set up. This paper presents an open-source R package (OpenMindat v1.0.0) to bridge the data highway, connecting users' overwhelming data needs, facilitating data-intensive query and access, unlocking novel insights, and enabling groundbreaking geoscience discoveries. The package was designed to be user-friendly and extensible. It exploits the capabilities of the Mindat API, including the data subjects of geomaterials (e.g., rocks, minerals, synonyms, variety, mixture, and commodity), localities,

and the IMA (International Mineralogical Association)-approved mineral list. In addition to providing functions for querying those data subjects, the package supports exporting data to various formats such as CSV, JSON-LD, and TTL. In applications, these functions only require minor coding and provide invaluable convenience for users with limited R environment experience. The package is open on GitHub under the MIT license and with detailed tutorial documentation. The field of mineralogy and many other geoscience disciplines are facing the opportunities enabled by open data. Various research topics

such as mineral network analysis, mineral association rule mining, mineral ecology, mineral evolution, and critical minerals have already benefited from Mindat's open data. We hope this R package will accelerate the process of those data-intensive studies and lead to more scientific discoveries.

## 1 Introduction

As machine learning and deep learning techniques thrive on their ability to discern intricate patterns and correlations, data-

driven discoveries in geosciences yield more exciting results (Reichstein et al., 2019; Bergen et al., 2019; Que et al., 2024). However, due to the complexity and multifaceted nature of Earth's processes, high-quality data are required to enable the capacity of machine learning and deep learning to make informed predictions and classifications across varying contexts (Chen et al., 2023). Therefore, open access to large and diverse datasets is imperative (Hossain et al., 2016), as this can unlock novel insights, enabling groundbreaking discoveries that elucidate the complexities underlying Earth's dynamic processes.



Regarding mineralogy, minerals provide many essential clues for exploring the complex geological history of the Earth and other planetary bodies (Prabhu et al., 2021). A rapidly growing volume of mineralogical and geochemical data resources are available for research, such as the IMA list of mineral species (rruff.info/ima) (Prabhu et al., 2023), Mindat (mindat.org) (Hazen et al., 2011), RRUFF (rruff.info) (Yang et al., 2011), EarthChem (earthchem.org) (Walker et al., 2005), the

Evolutionary System of Mineralogy Database (ESMD; odr.io/esmd) (Chiama et al., 2023), the Mineral Properties Database (odr.io/MPD) (Morrison et al., 2023), and the Astromaterials Data System (Astromat.org) (Chamberlain et al., 2021). Thanks to these big and expanding open datasets, new scientific topics such as mineral evolution (Hazen et al., 2008; Hazen et al., 2014), mineral ecology (Hazen et al., 2015), and mineral informatics (Prabhu et al., 2023) are emerging and developing dramatically. Among those data sources, the Mindat, a crowd-sourced database that started running in 2000, is now the world's

most widely used online database of mineralogical information. By August 2023, Mindat has recorded 5,960 minerals, 395,558 localities, 1,503,650 occurrences, and 1,291,077 photos, with a total data volume exceeding 25.8 TB (Ralph et al., 2022).

Although Mindat plays an increasingly significant role in scientific value and societal impacts, it still faces the challenges of infrastructure maintenance and development to meet the overwhelming data needs (Ma et al., 2024). In 2021 alone, the Mindat

website received 44,333,302 views from 10,148,136 unique visitors, and as of August 2023, the number of registered users reached 72,488. However, Mindat relies entirely on donations and sponsorships to maintain and develop the infrastructure to meet growing data volumes and needs. The undergoing project OpenMindat (Ma et al., 2024) aims to implement fully open access, machine-readable, and interoperable architecture for Mindat, making it an active data node in the geoscience cyberinfrastructure ecosystem.


Following the FAIR principles (i.e., findable, accessible, interoperable, and reusable), a roadmap of the OpenMindat project was laid out, including the technical approaches to upgrade and reuse existing data resources, tools, and infrastructure (Ma et al., 2024). In the Spring of 2023, the preliminary RESTful API (Richardson and Ruby, 2008) of Mindat was established, which registered users can access with an authorized API token. Although data can now be queried and accessed from the API

(https://api.mindat.org) (Zhang et al., 2024), a more friendly R or Python software package oriented to users' data needs, is still desired, as workflow platforms of R and Python such as R Markdown and Jupyter are now viral amongst geoscientists. Such an R or Python package of the Mindat API will make the data query and access much more accessible.

This paper presents our design and implementation of an R software package, OpenMindat v1.0.0, to meet users' needs for

quick and easy access to Mindat's open data. The package is open source for anyone to reuse, and we welcome feedback on improvement and extension. The remainder of this paper is organized as follows: Section 2 presents the technical architecture of the OpenMindat R package. Section 3 introduces the classes and functions in it. Section 4 presents a list of examples using this package, including records about geomaterials, localities, IMA minerals, and their outputs in different formats. Section 5



discusses the capabilities and limitations of the current R package and plans for future extension. Finally, Section 6 summarizes
the contributions of this study.

## 2 Technical Architecture of the OpenMindat R Package

The primary objective of the OpenMindat R package is to provide an implementation mechanism to translate users' data requirements into Mindat API requests. Mindat datasets, especially those made machine-readable through the Mindat RESTful API, are structured records stored in a relational MySQL database. The primary data subjects include mineral species (>5800),
alternative mineral names (>45000), localities (>390000), occurrence records (>1.2 million), photographs (>1.1 million), Mindat ID (>10.3 million), locality age (>5500), literature references (>13.5million), and meteoritics (1509 records including 449 petroleum categories). The API server manages HTTP requests for datasets. Currently, it provides a separate access endpoint for each data subject, and it distinguishes HTTP single-record requests and list requests as request query strings. Therefore, the key to building the R package is converting user data requirements into data request strings that the Mindat API
can handle.

Accordingly, we designed a technical architecture (**Figure 1**) oriented to users' data requirements. From the users' side, they do not need to care about the details of the request strings that will finally be sent to the API; they should operate on the functions in the R package and a token to connect to the Mindat API server. From our survey and interactions with geoscientists
in the past years, the users' data requirements fall into those categories: (1) Queries about geomaterials (i.e., mineral, rock, commodity, and other natural geological materials). Users need to query the geomaterials by their physical properties, such as density, hardness, color, refractive index, and crystal structure. Besides, users would also like to query datasets by chemical properties (e.g., element inclusion and exclusion states). Finally, Users need to filter geomaterial according to the entry type, such as synonym, variety, rock, mixture, mineral, and series. (2) Queries about localities. Mindat's localities record textual
addresses, coordinates, area boundaries, and other attributes. It is organized in a hierarchical structure. In most cases, users need to query the localities by their attributes, such as name, country, ID, description, and longitude/latitude. In addition, because of the hierarchical structure of localities, many users need to query localities by type, such as queries about whether a locality is at a bottom level or whether there are coordinate records. Moreover, studies of mineral evolution (Hazen and Ferry, 2010; Hystad et al., 2019) and the co-evolution of geosphere and biosphere (Hazen et al., 2014; Hazen and Morrison, 2020)
ignited the need to retrieve localities with geological ages. (3) Queries about IMA-approved minerals. The International Mineralogical Association (IMA) is an international group that promotes the science of mineralogy and standardizes the nomenclature of mineral species. Since the IMA mineral list is updated frequently, querying mineral information by the specific IAM status, such as A (approved), G (grandfathered), Rd (redefined), and Q (questionable), has also become one of the common needs.




In the implementation, the core module of the OpenMindat R package will first receive query functions and parameters, and then parse and distribute them to the corresponding endpoints. The API endpoint module will check its local cache, i.e., Mindat Cache, to see if it is already cached. If so, the data results module will return cached data and make a data frame to meet the data need. Otherwise, according to the query functions and parameters, the endpoint will assign query tasks to the specific sub-

endpoints, such as localities, geomaterials, or IMA minerals. The sub-endpoint will combine the query parameters and content to build a query URI that the Mindat API server can handle. When the query URI request is handled, the server will return the response data, and then the OpenMindat R package will extract the body of the response and parse the received raw data. The package can help users transform the retrieved data into various required formats, including CSV, TXT, and TTL, and support many applications, such as mineral association rule analysis (Morrison et al., 2023), mineral network analysis (Liu et al., 2018;

Morrison et al., 2020), and more.







**Figure 1: Technical architecture of the OpenMindat R package**

## 3 Capabilities and Usage

Following the designed technical architecture, we developed nine classes: Connection, Mindat Cache, Endpoint, Response Data Parser, Mindat Query, Geomaterials, Localities, IMA Minerals, and Data Maker. We implemented about 100 functions to ensure that the package could work accurately and efficiently. Table 1 lists the nine classes and briefly describes some of their tasks.

**Table 1: Capabilities of the OpenMindat R package (nine classes and some of their functions)**

| Class | Functions | Brief description |
|---|---|---|
| Connection | mindat_connection | Initialize the API Call. Setup the base URL, token of Mindat API, response page size per request, and the response format of API. |
| Mindat Cache | mindat_cache | Create a cache environment. |
| | mindat_cache_set | Set or allocate a cache by a given name. |
| | mindat_cache_get | Get cache value by name. If exist return the cached value, otherwise return NULL. |
| Endpoint | mindat_set_up_endpoints | Set up the base Mindat sub-endpoints (e.g., localities sub-endpoint, geomaterials sub-endpoint, etc.) according to the Mindat API server. |
| | mindat_uri_builder | Build a query URI based on the query dataset and parameters. |
| | mindat_setup | Set up a Mindat base URI, endpoints, and cache configuration information. |
| Response Data Parser | mindat_parse_raw_data | Parse the responded raw data (default is in JSON format) and then convert it to an R dataframe. If the raw data obtained is paged, the function will continue to request the next page of data until all data pages are obtained. |
| | mindat_extract_response_body | Check the response status for success. If so, text data is returned, otherwise an error is reported. |
| | mindat_make_data_frame | Convert the responded JSON data to an R dataframe. |
| | mindat_get_data_from_uri | Retrieve data from a query URI. |
| | mindat_build_querystring | Construct a query string based on query conditions. |
| Mindat Query | mindat_query | Basic function to query dataset from a specified endpoint. Entrance to all conditional queries. |




| | params_to_string | Parses arguments into strings so that the function can handle external conditions entered by the user. |
|---|---|---|
| | mindat_geomaterial(mindat_geo material_list) | Retrieve a (list of) geomaterial that matched the input conditions. |
| | mindat_localitiy(mindat_localities _list) | Retrieve a (list of) locality that matched the input conditions. |
| | mindat_mineral_ima(mindat_min eral_ima_list) | Retrieve a (list of) IMA mineral that matched the input conditions. |
| Data Maker | saveMindatDataAs | Save the data frame to file in a specified format. |
| | ConvertDF2JsonLD | Convert the retrieved data frame into a JSON-LD format string. |
| | ConvertDF2TTL | Convert the retrieved data frame into a TTL format string. |
| | geomaterials_contain_any_elems | Retrieve geomaterials that contain any of the specified elements. |
| | geomaterials_contain_all_elems | Retrieve geomaterials that contain all the specified elements. |
| | geomaterials_without_elems | Retrieve geomaterials that do not contain any of the specified elements. |
| | geomaterials_contain_only_elems | Retrieve geomaterials that only contain the specified elements. |
| | geomaterials_contain_all_and_wit hout_elems | Retrieve geomaterials that contain the specified included and excluded elements. |
| Geomaterials | geomaterials_cleavagetype | Retrieve geomaterials that matched the specified types. |
| | geomaterials_colour | Retrieve geomaterials that matched the specified colours. |
| | geomaterials_crystal_system | Retrieve geomaterials that matched the specified crystal system. |
| | geomaterials_bi_greater_than, geomaterials_bi_less_than | Retrieve geomaterials that had higher (lower) birefringence than the input value. |
| | geomaterials_dens_range | Retrieve geomaterials that matched the density within a given range. |
| | geomaterials_diapheny | Retrieve geomaterials that matched the given diapheny. |
| | localities_list_country | Retrieve the localities list that are found in a specified country. |
| Localities | localities_list_elems_inc, localities_list_elems_exc | Retrieve the localities that contain (or not contain) the given elements. |
| | localities_list_description | Retrieve the localities that contain the given description. |
| | minerals_ima_list | Retrieve the whole IMA mineral list. |
| IMA Minerals | minerals_ima_list_ima | Retrieve IMA mineral lists with given authorization statuses. |
| | minerals_ima_retrieve | Retrieve IMA mineral with given ID. |

From the software developers' point of view, these classes and functions follow a workflow sequence: (1) The Connection class helps establish the access configuration between the user and the Mindat API. Since the current API server only accepts



data access with authorization, it is necessary to register as a Mindat user and obtain a token to ensure a connection to the
server.    Users    can    apply    for    a    token    per    the    instructions    via    the    link:
https://www.mindat.org/a/how_to_get_my_mindat_api_key. Once the function "mindat_connection" is successfully executed,
other functions of the R package can run normally, and the connection configuration will be cached. (2) The Mindat Cache
class manages cached data, e.g., creation, release, acquisition, and more. Mindat data cache can reduce the number of
interactions between users and the API server and improve efficiency. (3) The Endpoint class is mainly responsible for
configuring and assigning sub-endpoints and constructing request URIs based on the query conditions. The sub-endpoints are
configured based on the Mindat API to handle different query datasets. For example, the geomaterials sub-endpoint is set up
to handle queries about records of geomaterials. The request URI to be sent to the Mindat API server for geomaterials can thus
be built according to the query conditions. (4) The Response Data Parser class processes the response data from Mindat API.
It can help check the response status, extract and parse the body of the response data, and convert it into a data frame of R. (5)
The Mindat Query is a comprehensive data retrieval class that requires the use of multiple classes mentioned above. It first
builds a query URI using an Endpoint instance, then sends a request to the API server via the URI, and finally parses the
response data using a Response Data Parser. (6) The Data Maker class is used for data conversion and output. It can convert
R data frames to required formats such as CSV, TXT, TTL, etc. (7) The Geomaterials class is one of the main dataset objects
supported by the Mindat API, and includes minerals, synonyms, varieties, mixtures, series, group lists, polytypes, rocks, and
commodities. The current geomaterial record contains 146 fields, including descriptions of physical properties, chemical
information, optical properties, crystal structure information, and more. (8) The Localities class is another vital dataset object
the Mindat API supports. It records 37 fields, including longitude, latitude, coordinate system, link, area, etc., which describe
the information of textual address, coordinate point position, locality type boundary polygon, and occurrences. (9) The IMA
Minerals class is mainly for retrieving and managing IMA-approved mineral species names, molecular formulas, authorization
status, and other attributes.

For the end users, those classes and functions can be applied flexibly to meet various data needs. For example, they support
retrieval of geomaterial records via single or combined query conditions, including filtering records by certain specified
chemical elements, hardness within a specified range, color characteristics, refractive index within a specified range, cut type
(e.g., imperfect/fair), crystal structure (e.g., amorphous), tenacity (e.g., brittle), and optical type (e.g., biaxial). They also
support retrieving localities records, e.g., a specific type (e.g., bottom level) of localities within a specified country (e.g.,
Sweden). More examples of this will be presented in the next section.

## 4 Examples and Results

To make the OpenMindat R package work adequately, some dependent packages, including "httr" and "jsonlite," should be
installed in the running environment. In addition, as mentioned above, an access token is required to establish a connection to



the Mindat API server. As shown in the following code, the "mindat_connection" function should be called to complete configuration initialization, and then all other functions can be called according to the user's data retrieval needs.

*R> library(httr)*
*R> library(jsonlite)*
*R> library (OpenMindat)*
*R> #Replace the following string "aa9c25fa95d8063908cb2bf186c9e79f" with your own Mindat token.*
*R> mindat_connection("aa9c25fa95d8063908cb2bf186c9e79f")*

The following examples demonstrate how to retrieve data through this OpenMindat R package, including the functions of the three main dataset entity classes: Geomaterials, Localities, and IMA Minerals.

## 4.1 Geomaterials

To illustrate the capabilities of retrieving geomaterial records, we chose some representative use cases. In terms of chemical elements of geomaterials, the package functions support data retrieval for relations including "contains any," "contains all,"
"contains only," "does not contain," "contains all but not," and "contains any but not." Some demo codes are as follows:

*R> geomaterials_contain_any_elems (c('Fe','S'))*
*R> geomaterials_contain_all_elems (c('Fe','S'))*
*R> geomaterials_contain_only_elems (c('Fe','S'))*
*R> geomaterials_not_contain_elems (c('Fe','S', 'O'), fields ="id,name,mindat_formula,elements")*
*R> geomaterials_contain_all_but_not_elems (c('Fe','S'), c('O'))*
*R>geomaterials_contain_any_but_not_elems(c('Fe','S'), c('O'))*

Results show that 10146 geomaterial records (148 fields) containing iron (Fe) or sulfur (S) elements were retrieved using the
function "geomaterials_contain_any_elems". The retrieved number of geomaterial records containing both Fe and S elements sharply dropped down to 1363 using the "geomaterials_contain_all_elems" function. Moreover, only 83 records of geomaterials that containing only Fe and S elements (no other elements) were retrieved by using the "geomaterials_contain_only_elems" function. Besides, 31919 records of geomaterials that did not contain elements Fe, S, and O (Oxygen) (the fields were filtered by id, name, mindat_formula, and elements) were retrieved by using the
"geomaterials_not_contain_elems" function. Lastly, 808 records of geomaterials containing Fe and S, but not O elements can be retrieved using the "geomaterials_contain_all_but_not_elems" function. **Table 2** shows the head 3 of geomaterial records in those examples.





**Table 2: Results of geomaterial records retrieved by chemical elements**

| Function name | Head 3 records of some selected fields | | |
|---|---|---|---|
| | name | mindat_formula | elements |
| *geomaterials_contain_any_elems* | Ach√°valite | FeSe | -Fe-Se- |
| | Actinolite | ◻Ca$_2$(Mg$_{4.5-2.5}$Fe$_{0.5-2.5}$)Si$_8$O$_{22}$(OH)$_2$ | -Ca-Fe-Mg-Si-O-H- |
| | Aegirine | NaFe$^{3+}$Si$_2$O$_6$ | -Fe-Na-Si-O- |
| *geomaterials_contain_all_elems* | Aluminocopiapite | Al$_{2/3}$Fe$^{3+}$$_4$(SO$_4$)$_6$(OH)$_2$·20H$_2$O | -Al-Fe-O-S-H- |
| | Alloclasite | Co$_{1-x}$Fe$_x$AsS | -As-Co-Fe-S- |
| | Amarantite | Fe$^{3+}$$_2$(SO$_4$)$_2$O·7H$_2$O | -Fe-O-S-H- |
| *geomaterials_contain_only_elems* | Gold-bearing Pyrite | FeS$_2$ | -Fe-S- |
| | Greigite | Fe$^{2+}$Fe$^{3+}$$_2$S$_4$ | -Fe-S- |
| | Mackinawite | FeS | -Fe-S- |
| *geomaterials_not_contain_elems* | Abelsonite | Ni(C$_{31}$H$_{32}$N$_4$) | -Ni-N-C-H- |
| | Aluminium | Al | -Al- |
| | Algodonite | (Cu$_{1-x}$As$_x$) | -As-Cu- |
| *geomaterials_contain_all_but_not_elems* | Alloclasite | Co$_{1-x}$Fe$_x$AsS | -As-Co-Fe-S- |
| | Bismuth-bearing Tetrahedrite | Cu$_6$(Cu$_4$(Fe/Zn)$_2$)(Sb,Bi)$_4$S$_{12}$S | -Bi-Cu-Fe-Sb-Zn-S- |
| | Argentopentlandite | Ag(Fe,Ni)$_8$S$_8$ | -Ag-Fe-Ni-S- |
| *geomaterials_contain_any_but_not_elems* | Achávalite | FeSe | -Fe-Se- |
| | Alloclasite | Co$_{1-x}$Fe$_x$AsS | -As-Co-Fe-S- |
| | Bismuth-bearing Tetrahedrite | Cu$_6$(Cu$_4$(Fe/Zn)$_2$)(Sb,Bi)$_4$S$_{12}$S | -Bi-Cu-Fe-Sb-Zn-S- |

Code and results shared on GitHub:
https://github.com/quexiang/OpenMindat/blob/main/notebook/Retriev_Geomaterials_by_elements.ipynb



In addition to chemical elements, the package also supports geomaterial dataset retrieval by using physical properties, including density, hardness, birefringence, optical 2v, crystal system, fracture type, color, streak, diaphaneity, lustre type, optical sign,
optical type, poly type, cleavage type, tenacity, and more. Some demo codes are as follows:

*R> geomaterials_hardness_gt (9)*

*R> geomaterials_hardness_lt (1)*

*R> geomaterials_hardness_range(3,3.5)*

*R> geomaterials_dens_range(3,3.2)*

*R> geomaterials_optical2v_range (9,10)*

**Table 3** shows the head 3 of retrieved geomaterial records that the numerical physical properties meet the given conditions. For example, the hardness of geomaterials in the Mindat database, ranging from 0 to 10, usually refers to the Mohs scale (Broz
et al., 2006). The "geomaterials_hardness_gt" and "geomaterials_hardness_lt" functions retrieve geomaterial records with hardness higher and lower than a given value, respectively. The "geomaterials_hardness_range" function retrieves the records whose density is within a specific interval. In addition to hardness, records of similar physical properties (e.g., density or optical 2v) can also be retrieved by calling the corresponding functions such as "geomaterials_dens_range" and "geomaterials_optical2v_range".


Table 3: Results of geomaterial records retrieved by using physical properties in numeric forms

| Function name | Head 3 records of some selected fields | | |
| --- | --- | --- | --- |
| | name | min_value | max_value |
| *geomaterials_hardness_gt* | Bahianite | 9 | 9 |
| | Bromellite | 9 | 9 |
| | Chromium | 9 | 9 |
| *geomaterials_hardness_lt* | Acetamide | 1 | 1.5 |
| | Aliettite | 1 | 2 |
| | Aluminite | 1 | 2 |
| *geomaterials_hardness_ra nge* | Abelsonite | 2 | 3 |
| | Abernathyite | 2.5 | 3 |
| | Acuminite | 3.5 | 3.5 |
| *geomaterials_dens_range* | Actinolite | 3.03 | 3.24 |
| | Akrochordite | 3.194 | 3.35 |
| | Amblygonite | 3.04 | 3.11 |
| | Autunite | 10 | 53 |





| | | | |
|---|---|---|---|
| ***geomaterials_optical2v_ra*** | Bario-orthojoaquinite | 10 | 15 |
| ***nge*** | Beidellite | 9 | 16 |

Code and results shared on GitHub:
https://github.com/quexiang/OpenMindat/blob/main/notebook/Retrieve_Geomaterials_by_physical_prop_1.ipynb

The geomaterial records can also be retrieved using special symbols and strings representing different physical features. The following codes show some examples:

*R> geomaterials_crystal_system(c("Icosahedral"))*

*R> geomaterials_fracturetype(c("Step-Like"))*

*R> geomaterials_colour(c("bright blue"))*

*R> geomaterials_streak("orange")*

*R> geomaterials_diapheny("Transparent")*

*R> geomaterials_lustretype(c("Adamantine"))*

*R> geomaterials_opticalsign("-")*

*R> geomaterials_polytypeof(0)*

*R> geomaterials_cleavagetype(c("Poor/Indistinct"))*

**Table 4** shows part of the results retrieved by symbols or strings with actual physical meaning. The last column is the field that matches the input string or symbols (except for the field of "commentcrystal"), and the remaining columns are some
related fields. Results only list the head 3 records that meet the conditions (all those with less than 3 records are also listed), showing that these implemented functions can run accurately and efficiently.

**Table 4: Results of geomaterial records retrieved by using physical properties in symbol and string forms**

| *Function Name* | *Head 3 records of some selected fields* | | | |
|---|---|---|---|---|
| | *name* | *elements* | *csystem* | *commentcrystal* |
| ***geomateri als_crysta l_system*** | Icosahsedrite | -Al-Cu-Fe- | Icosahedral | The structure is not reducible to a single three-dimensional unit cell, so neither cell parameters nor Z can be given. The X-ray powder pattern was indexed on the basis of six integer indices, as conventionally used with quasicrystals, where the lattice parameter (in six-dimensional notation) is measured to be a6D = 12.64 √Ö, with probable space group Fm-3-5. |
| | Decagonite | -Al-Fe-Ni- | Icosahedral | P105/mmc. |



| | name | elements | csystem | | |
|---|---|---|---|---|---|
| Unnamed (Mn-Si-Cr-Al-Ni Quasicrystal) | | -Al-Cr-Mn-Ni-Si- | Icosahedral | | |

| | name | elements | csystem | cleavagetype | fracturetype |
|---|---|---|---|---|---|
| geomaterials_fracturetype | Bytownite | -Al-Ca-Na-Si-O- | Triclinic | Perfect | Step-Like |
| | Clinoatacamite | -Cl-Cu-O-H- | Monoclinic | Perfect | Step-Like |
| | Daomanite | -As-Cu-Pt-S- | Orthorhombic | Distinct/Good | Irregular/Uneven,Step-Like |

| | name | elements | csystem | colour | |
|---|---|---|---|---|---|
| geomaterials_colour | Astrocyanite-(Ce) | -Ce-Cu-La-Nd-O-C-H-U- | Hexagonal | Bright blue | |
| | Chlorothionite | -Cl-Cu-O-K-S- | Orthorhombic | Bright blue. | |
| | Lautenthalite | -Cu-Pb-O-S-H- | Monoclinic | Green, bright blue | |

| | name | elements | csystem | colour | streak |
|---|---|---|---|---|---|
| geomaterials_streak | Alacránite | -As-S- | Monoclinic | Orange to pale gray with rose-yellow internal reflections | Yellow-orange |
| | Berzeliite | -As-Ca-Mg-Na-O- | Isometric | Yellow, Orange, colorless, brownish-orange; colorless to orange in transmitted light. | Nearly white to yellow-orange |
| | Cassedanneite | -Cr-Pb-O-H-V- | Monoclinic | Orange-red | Yellow-orange |

| | name | elements | csystem | colour | diapheny |
|---|---|---|---|---|---|
| geomaterials_diapheny | Abenakiite-(Ce) | -Ce-Na-Si-O-P-C-S- | Trigonal | Pale brown | Transparent |
| | Abernathyite | -As-O-K-H-U- | Tetragonal | yellow | Transparent |
| | Abhurite | -Cl-Sn-O-H- | Trigonal | Colourless | Transparent |

| | name | elements | csystem | lustre | lustretype |
|---|---|---|---|---|---|
| geomaterials_lustretype | Aerugite | -As-Ni-O- | Trigonal | | Sub-Adamantine,Sub-Vitreous,Resinous |
| | Annabergite | -As-Ni-O-H- | Monoclinic | Weakly adamantine, vitreous, earthy when powdery. | Sub-Adamantine,Sub-Vitreous,Pearly,Earthy |
| | Ardennite-(As) | -Al-As-Mg-Mn-Si-O-H- | Orthorhombic | | Sub-Adamantine |





| geomaterials_optica lsign | name | elements | colour | csystem | opticalsign |
|---|---|---|---|---|---|
| | Abenakiite-(Ce) | -Ce-Na-Si-O-P-C-S- | Pale brown | Trigonal | - |
| | Abernathyite | -As-O-K-H-U- | yellow | Tetragonal | - |
| | Acetamide | -N-O-C-H- | Colourless, grey | Trigonal | - |

| geomaterials_polyty peof | name | elements | csystem | opticalbireflectance | polytypeof |
|---|---|---|---|---|---|
| | Barikaite | -Ag-As-Pb-Sb-S- | Monoclinic | distinct bireflectance in grey tones, | 0 |
| | Pašavaite | -Pb-Pd-Te- | Orthorhombic | strong bireflectance | 0 |
| | Batisivite | -Ba-Si-Ti-O-V- | Triclinic | weakly bireflected | 0 |

| geomaterials_cleava getype | name | elements | csystem | cleavage | cleavagetype |
|---|---|---|---|---|---|
| | Abelsonite | -Ni-N-C-H- | Triclinic | Probable on <mi>{11_1}</mi>. | Poor/Indistinct |
| | Abenakiite-(Ce) | -Ce-Na-Si-O-P-C-S- | Trigonal | {0001} | Poor/Indistinct |
| | Aikinite | -Bi-Cu-Pb-S- | Orthorhombic | on {010} | Poor/Indistinct |

Code and results shared on GitHub:

https://github.com/quexiang/OpenMindat/blob/main/notebook/Retrieve_Geomaterials_by_physical_prop_2.ipynb

Besides the above-mentioned, geomaterial records can be retrieved using wildcard names, specifying non-null fields of interest (whether the specified fields are empty), Mindat ID, mineral varieties, and more. These functions can sometimes greatly facilitate data retrieval needs. Some demo codes are as follows:


*R> geomaterials_search_name("Quartz")*

*R> geomaterials_name("qu_rtz")*

*R> geomaterials_name("_u_r_z")*

*R> geomaterials_name("qu*")*

*R> geomaterials_field_exists("meteoritical_code",TRUE)*

*R> mindat_geomaterial(id=3337)*

*R> geomaterials_varietyof(3337)*

*R> geomaterials_entrytype(c('1'))*

*R> mindat_geomaterial_list(ids = c('3','3337'))*






**Table 5** shows the geomaterial records retrieved by wildcard name, non-null fields, Mindat ID, and entry type. A brief description of the function and the meaning of the arguments are also included.

<p style="text-align:center">Table 5: Results of geomaterial records retrieved by wildcard names, non-null fields, and entry type</p>

| *Function name & its brief description* | *Head 3 records of some selected fields* | | |
|---|---|---|---|
| ***geomaterials_search_name*** | *id* | *name* | |
| **Input**: "Quartz", a full name of geomaterials | 1877 | \Herkimer-style\" Quartz" | |
| **Output**: The records that match or contain the input full name | | | |
| | 6124 | α-Quartz | |
| | 10773 | Alpha-Quartz | |
| ***geomaterials_name*** | *id* | *name* | |
| **Input**: "qu_rtz". **Output**: records of geomaterials whose names match the input 6-character wildcard names, where the third character were arbitrary. | 3337 | Quartz | |
| | 6747 | Quertz | |
| **Input**: "_u_r_z". **Output**: records of geomaterials whose names match the input 6-character wildcard names, where the first, third, and fifth characters were arbitrary. | 3337 | Quartz | |
| | 6747 | Quertz | |
| **Input**: "qu*". **Output**: records of geomaterials whose names had the first two characters matched the input two characters (i.e., q and u). | 3335 | Quadridavyne | |
| | 3336 | Quadruphite | |
| | 3337 | Quartz | |
| ***geomaterials_field_exists*** | *id* | *name* | *meteoritical_code* |
| **Input**: "meteoritical_code", a field of geomaterials. TRUE, Boolean value. **Output**: records of geomaterials that had non-null values of the field "meteoritical_code". | 11263 | Lodranite meteorite | Lodranite |
| | 49515 | Chondrite meteorite | Chondrite-uncl |
| | 49517 | Chondrite-fusion crust meteorite | Chondrite-fusion crust |
| ***mindat_geomaterial*** | *id* | *name* | |
| **Input**: id=3337 or 3337, specify a mindat id.**Output**: records of geomaterials that had the same mindat id as the input id. | 3337 | Quartz | |
| | 3337 | Quartz | |
| ***geomaterials_varietyof*** | *id* | *name* | *varietyof* |
| **Input**:3337, which is the mindat id of Quartz. **Output**: records of geomaterials that were varieties of Quartz. | 198 | Amethyst | 3337 |
| | 398 | Star Quartz | 3337 |
| | 436 | Aventurine | 3337 |
| ***geomaterials_entrytype*** | *id* | *name* | *entrytype* |




| | | |
|---|---|---|
| **Input**: 2, an integer.**Output**: records of geomaterials that match | 8 | Absite | 2 |
| the input entry type, which can be one value of below:0: mineral; | 12 | Acarbodavyne | 2 |
| 1: synonym; 2: variety;3: mixture; 4: series; 5: grouplist; 6: | 22 | Adamsite | 2 |
| polytype;7: rock; 8: commodity | | | |

| mindat_geomaterial_list | id | longid | name |
|---|---|---|---|
| **Input**: ids = c ('3','3337'), a list of mindat IDs. **Output**: records | 3 | 1:1:3:3 | Abernathyite |
| of geomaterials with the same Mindat ID as the input ID list. | 3337 | 1:1:3337:0 | Quartz |

Code and results shared on GitHub:
https://github.com/quexiang/OpenMindat/blob/main/notebook/Retrieve_Geomaterials_by_wildcar_names.ipynb

Lastly, the package can also retrieve geomaterial datasets based on combined conditions. For example, if we would like to retrieve an IMA-approved minerals record containing elements Li and O, Mohs hardness between 5.8 and 6, and in the triclinic
crystal structure, the following function can be used:

*R> geomaterials_contain_all_elems(c('Li','O'), hardness_min = 5.8, hardness_max = 6, crystal_system = "Triclinic",ima_status = "APPROVED",entrytype = 0)*

**Table 6** shows the retrieved records matching those conditions. Users can choose a related function as the main query function (e.g., the "geomaterials_contain_all_elems" function). Then additional conditions can be added to the main query function according to the actual data needs and conditions. The field names in the additional conditions can be seen in the online Mindat API document (https://api.mindat.org/schema/redoc).

**Table 6: Results of geomaterial records retrieved by combined conditions**

| Function | | | | | | | | |
|---|---|---|---|---|---|---|---|---|
| Name & input description | id | name | elements | hmin | hmax | csystem | ima_status | entrytype |
| *geomaterials_contain_all_elems* | 189 | Amblygonite | "Al", "Li", "O", "P", "F" | 5.5 | 6 | Triclinic | "APPROVED", "GRANDFATHERED" | 0 |
| **Input:** c('Li','O'), contain Li and O elements. hardness_min = 5.8 and hardness_max = 6, crystal_system = "Triclinic", crystal structure is Triclinic. ima_status = "APPROVED", | 670 | Bikitaite | "Al", "Li", "Si", "O", "H" | 6.0 | 6 | Triclinic | "APPROVED", "GRANDFATHERED" | 0 |
| IMA approved. entrytype = 0, entrytype is mineral. **Output:**Geomaterials records that matched the combined conditions. | 2417 | Lithiomarsturite | "Ca", "Li", "Mn", "Si", "O", "H" | 6.0 | 6 | Triclinic | "APPROVED" | 0 |

Code and results shared on GitHub:
https://github.com/quexiang/OpenMindat/blob/main/notebook/Retrieve_Geomaterials_by_combined_conds.ipynb





**4.2 Localities**

The current package supports Mindat locality dataset retrieval by using country name, Mindat ID, element inclusion
relationship, and other conditions. Some example codes are given below:

*R> localities_list_country ("China")*

*R> localities_retrieve_id(id = 22)*

*R> localities_list_description("volcano")*


As shown in **Table 7**, users can easily retrieve the locality records by a given country through the function
"localities_list_country". The current API will return Mindat locality records containing 37 fields, among which commonly
used fields including id, country, txt (text), latitude, longitude, element, age, level, etc. Mindat locality follows a specific
hierarchical structure and naming rules; users can refer to this link: https://www.mindat.org/a/localityhierarchies. The
"latitude" and "longitude" fields record the coordinates of a locality, and the "txt" field records a text string of locations that
follows the hierarchy described above. The number of locality levels reflects the level of detail of the address. The larger the
value, the more specific the address information is. Thus, 0 is the top level and usually represents a country, a region, or a
tectonic plate. The field "description_short" briefly introduces the locality. Users can retrieve records of terms contained in
the "description_short" field using the "localities_list_description" function. The field "element" records the elements of the
locality, that is, the list of elements in all the mineral species found at this locality. Users can quickly obtain records of
"contains", "does not contain", and "contains but does not contain" chemical element relationships through the
"localities_list_elems_inc", "localities_list_elems_exc", and "localities_list_elems_inc_exc" functions respectively. The
"age" field of Mindat Locality records the "age_id" of a locality age. In addition, although the current locality record does not
include a mineral list, we are aware of the needs of many users. We are planning to implement it in a future extension.


**Table 7: Locality records retrieved by different inputting conditions**

| Function name & description | | | Head 3 records of some selected fields | | | | |
|---|---|---|---|---|---|---|---|
| *localities_list_country* | *id* | *country* | *element* | *latitude* | *longitude* | *level* | *Txt* |
| **Input**: "China", a country name. **Output**: The localities records occur or are within the specified country. | 693 | China | -Ag-S-Fe-Se-Ca-Mg-Si-O-H-As-Zn-Al-K-Na-Ti-C-Ce-Nd-Y-F-Bi-Cu-Pb-Cl-Hg-Mn-Th-Te-La-Sb-Co-Ba-Li-P-Sr-Ru-Ni-Au-Be-Pd-Ge-Nb-Sn-U-Pt-Tl-Zr-B-Mo-Ta-Rh-V-Br-N-Cr-Ir-In-Os-W-Cs-Cd-Hf-I-Sc-Ga-Rb- | 0 | 0 | 0 | China |



| | id | country | element | latitude | longitude | level | Txt |
|---|---|---|---|---|---|---|---|
| A total of 10026 records returned. | 694 | China | -Ca-Fe-Mg-Si-O-Al-Ti-K-F-H-C-Cl-P-Cr-Cu-Na-Ni-S-Zn- | 33.833333333 | 115.8333333 | 5 | Boxian meteorite, Xiaoyanzhuang, Qiaocheng District, Bozhou, Anhui, China |
| | 695 | China | -Ag-S-Ca-Fe-Mg-Si-O-H-Mn-Al-Na-Bi-Pb-Te-Sb-C-As-Ni-Cl-Cu-Ba-Ti-K-F-B-Be-Sn-Co-Sr-Zn-Cr-Hg-U-Au-Cd-Mo-P-W-Li-Ta-Zr- | 0 | 0 | 2 | Haixi Mongol and Tibetan Autonomous Prefecture, Qinghai, China |

| localities_retrieve_id | id | country | element | latitude | longitude | level | Txt |
|---|---|---|---|---|---|---|---|
| **Input**: "22", Mindat Id. **Output**: the locality matches the input ID. | 22 | Algeria | -As-Ba-Fe-O-H- | 36.521388889 | 7.295 | 4 | Djebel Debar, Roknia, Hammam Debagh District, Guelma Province, Algeria |

| localities_list_description | id | country | element | latitude | longitude | level | Description_short |
|---|---|---|---|---|---|---|---|
| **Input**: "volcano", an enter terms that must be contained in the descriptive text (i.e., field of description_short). **Output**: records of localities that matched the input string. A total of 1729 records return. | 38 | Antarctica | -Al-Ca-Fe-Mg-Na-Si-O-K-S-H-Cl-C-Ti-F-P-Au-Mn- | -77.526088838 | 167.0800781 | 4 | Ross Island is an island formed by four volcanoes in the Ross Sea near the continent of Antarctica, off the coast of Victoria Land in McMurdo Sound. Ross Island lies within the boundaries of Ross Dependency, an area of Antarctica claimed by New Zealand. |
| | 113 | Australia | -Fe-Na-Si-O-Ti-Al-H-Ca-Mg-Cl-Zr-K- | -31.315 | 149.185 | 4 | The shield volcano that made the Warrumbungle ranges was active about 13-17 million years ago. The volcano had a roughly circular outcrop 50 km in diameter, but is now heavily eroded and dissected, with prominent sub-volcanic dykes (Duggan 1989). |
| | 120 | Australia | -Ca-Fe-Mg-Si-O-H-Al-Na-K-Be-Cl-Cu-P-U-C-Ba-S-Ti-F-Bi-V-Sn-Li-B-Nb-Y-Zn-Pb-Au-Ag-Mo-Mn-W-Ta-Th-Zr- | -22.986666667 | 134.9208333 | 3 | Harts Range is about 190 kilometres by dirt road (Stuart and Plenty Hwys) north-east of Alice Springs. Areas most accessible and interesting tend to be along the northern limits of the ranges. The eastern and southern sections can be accessed by a trac... |

Note: code and results: https://github.com/quexiang/OpenMindat/blob/main/notebook/Retrieve_Localities_by_desc.ipynb



In addition to the above locality record retrieval functions, the locality age, status, type, and other fields can also be used to
filter records from the Mindat API. However, there are some limitations due to the safety protection set on the API server to
reduce extremely heavy data outputs. Our R package currently only implements some functions related to the locality age. For
example:

*R> localities_list_elems_exc(c("H", "O", "Si", "Al", "Fe", "Ca", "Na", "K", "P", "C", "Mn", "F", "Mg", "S"))*
*R> localities_list_elems_inc(c("Dy"))*
*R> localities_list_elems_inc_exc(c("Dy"), c("Li"))*
*R> locality_age_list()*
*R>locality_age(id = 60)*

As shown in **Table 8**, the locality age records can be retrieved by the "locality_age_list" function, and the results showed that
the geological time interval of a locality was recorded via the "age_mav" and "age_ma2v" fields. The "age_id" is the unique
identifier associated with the locality and locality age, i.e., if a locality has its corresponding locality age, then the "age_id" of
locality age will be recorded in the "age" field. The field "agemethod" recorded the method used to obtain the estimated
geological time.


**Table 8: Results of locality records retrieved by status, age, type, and a few other fields**

| Function name & its brief description | Head 3 records of some selected fields | | | | | |
|---|---|---|---|---|---|---|
| *localities_list_elems_exc* | *country* | *elements* | *latitude* | *longitude* | *level* | *Txt* |
| **Input**: c ("H", "O", "Si", "Al", "Fe", "Ca", "Na", "K", "P", "C", "Mn", "F", "Mg", "S"). **Output**: records of localities that do not contain the input elements. A total of 223622 records returned. | Afghanistan | - | 0 | 0 | | Mohammad Agha District, Logar, Afghanistan |
| | | | | | 2 | |
| | Australia | -Au- | -30.23207267 | 151.8779073 | 4 | Aberfoyle River deep lead, Aberfoyle River, Clarke Co., New South Wales, Australia |
| | Australia | -Ag-Cl- | -36.5889328 | 143.244155 | 4 | Silver Mines (Silver Reef), St Arnaud, Northern Grampians Shire, Victoria, Australia |
| *localities_list_elems_inc*<br><br>**Input**: c("Dy"), elements that the localities contain. | Germany | -As-O-K-H-U-Ag-S-Ca-Fe-Mg-Si-Zn-Al-Na-Cu-Nd-Y-Ce-Dy-La-Bi-Pb-Cl-Sb-Co-N-Ti-C-Ni-Be-Sr-Ba-Ge-F-Mn-Au-P-Zr-Se- | 0 | 0 | 1 | Baden-Württemberg, Germany |



| | | | | | | |
|---|---|---|---|---|---|---|
| **Output**: records of localities that include the input elements. 9 records returned. | Germany | Br-V-Sn-Nb-Cr-Hg-Li-W-B-Th-Cd-Tl-I-Mo--Ag-S-As-Fe-Al-O-F-H-U-Ca-P-Ba-Na-Si-Bi-Cu-C-Y-Pb-Se-K-Li-Mg-Au-Co-Nd-N-Ni-Mn-Zn-W-Dy- | 47.838 88889 | 8.0488 88889 | 0 | Krunkelbach Valley Uranium deposit, Menzenschwand, St Blasien, Waldshut, Freiburg Region, Baden-Württemberg, Germany |
| | Germany | -As-O-K-H-U-Ag-S-Zn-Cu-Nd-Y-Ce-Dy-Bi-Pb-Co-Fe-Al-Si-Ti-Ca-C-Ni-F-Mn-Ba-Mg-Cl-Hg-Na-P-Sb-Mo-N- | 48.338 10891 | 8.3434 29565 | 5 | Wittichen, Schenkenzell, Rottweil, Freiburg Region, Baden-Württemberg, Germany |
| ***localities_list_elems_inc_exc***<br>**Input**: c("Dy"), elements that the localities contain. c("Li"), elements that the localities does not contain.<br>**Output**: records of localities that match the input condition.3 records returned. | Germany | -As-O-K-H-U-Ag-S-Zn-Cu-Nd-Y-Ce-Dy-Bi-Pb-Co-Fe-Al-Si-Ti-Ca-C-Ni-F-Mn-Ba-Mg-Cl-Hg-Na-P-Sb-Mo-N- | 48.338 10891 | 8.3434 29565 | 5 | Wittichen, Schenkenzell, Rottweil, Freiburg Region, Baden-Württemberg, Germany |
| | Germany | -As-O-K-H-U-Ag-S-Ca-Fe-Mg-Si-Zn-Cu-Nd-Y-Ce-Dy-Bi-Pb-Co-Al-Ti-C-Ni-F-Mn-Ba-Cl-Hg-Na-P-Sb-Mo- | 0 | 0 | 4 | Schenkenzell, Rottweil, Freiburg Region, Baden-Württemberg, Germany |
| | Germany | -As-O-K-H-U-Ag-S-Ca-Fe-Mg-Si-Zn-Cu-Nd-Y-Ce-Dy-Bi-Pb-Co-Al-Ti-C-Ni-F-Mn-Ba-Cl-Be-Hg-Na-P-Sb-B-Mo- | 0 | 0 | 3 | Rottweil, Freiburg Region, Baden-Württemberg, Germany |

| ***locality_age_list*** | ***age_id*** | ***age_mav*** | ***age_ma2v*** | ***agemethod*** |
|---|---|---|---|---|
| | 3 | 170.3 | 157.3 | K/Ar |
| | 17 | 4574.7 | 4574.7 | Pb-Pb isochrons |
| | 36 | 590 | 590 | Pre-1977 K-Ar, Ar-Ar and Rb-Sr ages recalculated using the decay constants of Steiger and Jager (1977) |
| ***locality_age*** | 60 | 717.4 | 660 | Re-Os |

Code and results shared on GitHub: https://github.com/quexiang/OpenMindat/blob/main/notebook/Retrieve_Localities_by_elems.ipynb

### 4.3 IMA minerals

The R package makes retrieving an IMA-approved mineral list relatively easy. The code below shows how to retrieve IMA
minerals in a whole list, by their status, or by ID. Table 9 lists a few fields in the results. The field "type locality" denotes
where the original material came from for the formal definition of the mineral species.





R> minerals_ima_list()

R> minerals_ima_list_ima(1)

R> minerals_ima_retrieve(2)

**Table 9: Results of IMA minerals records retrieved by different constraint conditions**

| Function name | Head 3 records of some selected fields | | | | |
|---|---|---|---|---|---|
| *minerals_ima_list* | *name* | *Type_localities* | *ima_formula* | *ima_status* | |
| | Abelsonite | 39262 | $NiC_{31}H_{32}N_4$ | APPROVED | |
| | Abenakiite-(Ce) | 599 | $Na_{26}Ce_6(Si_6O_{18})(PO_4)_6(CO_3)_6(SO_2)O$ | APPROVED | |
| | Abernathyite | 4145 | $K(UO_2)(AsO_4) \cdot 3H_2O$ | c("APPROVED", "GRANDFATHERED") | |
| *minerals_ima_list_ima(1)* | *name* | *Type_localities* | *mindat_formula_note* | *ima_status* | *ima_formula* |
| | Paramolybdomenite | 333762 | $PbSeO_3$ | c("APPROVED", "PENDING_PUBLICATION") | |
| | Mckelveyite-(Nd) | 435543 | $NaCaBa_3Nd(CO_3)_6 \cdot 3H_2O$ | c("APPROVED", "PENDING_PUBLICATION") | $NaCaBa_3Nd(CO_3)_6 \cdot nH_2O$ |
| | Naalasite | 190910 | $NaAl(AsO_3OH)_2 \cdot H_2O$ | c("APPROVED", "PENDING_PUBLICATION") | |

| Function name | Head 3 records of some selected fields | | | | | |
|---|---|---|---|---|---|---|
| *minerals_ima_retrieve(2)* | *id* | *name* | *mindat_formula* | *ima_status* | *elements* | *sigelements* |
| | 2 | Abenakiite-(Ce) | $Na_{26}Ce_6(Si_6O_{18})(PO_4)_6(CO_3)_6(SO_2)O$ | APPROVED | Ce | Ce |
| | 2 | Abenakiite-(Ce) | $Na_{26}Ce_6(Si_6O_{18})(PO_4)_6(CO_3)_6(SO_2)O$ | APPROVED | Na | Na |





| 2 | Abenak iite- (Ce) | Na$_{26}$Ce$_6$(Si$_6$O$_{18}$)(PO$_4$)$_6$(CO$_3$)$_6$(SO$_2$)O | APPROV ED | Si | Si |
|---|---|---|---|---|---|

Code and results shared on GitHub: https://github.com/quexiang/OpenMindat/blob/main/notebook/IMA_minerals.ipynb

### 4.4 Output files in different formats

The current R package supports users in outputing their retrieved data in various formats, including CSV, JSON, TXT, JSON-LD, and TTL. The function "saveMindatDataAs" will identify the suffix of the output file name and then convert the R data frame into the corresponding format. For data conversion to the JSON-LD and TTL formats, the two Excel template files (i.e., OpenMindat_Schema_JSON-LD.xlsx and OpenMindat_Schema_TTL.xlsx) were required, which can be accessed via https://github.com/quexiang/OpenMindat/tree/main/inst/extdata. Users can configure their settings in the Excel template to

customize files that meet their own needs for the output. Here, we take the JSON-LD template as an example to briefly introduce its basic settings (similar template settings in TTL format). Two Excel sheets are in the template file; the first is about context settings. **Table 10** shows the names of all schemas and how their corresponding URLs are configured. The other concerns are field settings, as shown in Table 11, where "fields" record the field names that need output corresponding to the Mindat API. In this sheet, the "ref_fields" records the output field name list of JSON-LD, "context_name" records all schema

names corresponding to the field, and "type" records the type of schema to which the field belongs. All the values of the three fields are in the form of a list, separated by commas. Besides, the "ref_field_num" indicates which name is to be output in JSON-LD (e.g., 1 represents the name before the first comma of "ref_fields").

**Table 10: Context settings of the JSON-LD template**

| context_name | context_url |
|---|---|
| mindat | https://mindat.org/ |
| schema | https://schema.org/ |
| gsog | https://w3id.org/gso/geology/ |


**Table 11: Field settings of the JSON-LD template**

| fields | ref_fields | context_name | type | ref_field_num |
|---|---|---|---|---|
| id | mindat:id, , | mindat,schema,gsog | mindat:Geomaterials,schema:Dataset,gsog:Mineral_Material | 1 |
| longid | identifier, , | mindat,schema,gsog | mindat:Geomaterials,schema:Dataset,gsog:Mineral_Material | 1 |
| name | mindat:name, , | mindat,schema,gsog | mindat:Geomaterials,schema:Dataset,gsog:Mineral_Material | 1 |
| ima_formula | mindat:ima_formula, , | mindat,schema,gsog | mindat:Geomaterials,schema:Dataset,gsog:Mineral_Material | 1 |

The full JSON-LD template share on GitHub:
https://github.com/quexiang/OpenMindat/blob/main/inst/extdata/OpenMindat_Schema_JSON-LD.xlsx



According to the above configuration, we can obtain the file shown in **Table 12** by executing the following code:

*R> library(readxl)*

*R> saveMindatDataAs(geomaterials_hardness_gt(9.8,fields = "id,longid,name ,ima_formula"),"df_geomaterials.jsonld ")*

**Table 12: Output file in JSON-LD format**

| *df_geomaterials.jsonld* |
|---|

```
{
 "@context": {
         "mindat":"https://mindat.org/",
         "schema":"https://schema.org/",
          "gsog":"https://w3id.org/gso/geology/"},
 "@graph": [{"@type": ["mindat:Geomaterials " ,  "schema:Dataset " ,  gsog:Mineral_Material "] ,
                 " mindat:id ":" 1282 ",
                 " identifier ":" 1:1:1282:5 ",
                 " mindat:name ":" Diamond ",
                 " mindat:ima_formula ":" C "
         }
         , {"@type": ["mindat:Geomaterials " ,  "schema:Dataset " ,  gsog:Mineral_Material "] ,
                 " mindat:id ":" 43792 ",
                 " identifier ":" 1:1:43792:7 ",
                 " mindat:name ":" Qingsongite ",
                 " mindat:ima_formula ":" BN "
         } ,
         {"@type": ["mindat:Geomaterials " ,  "schema:Dataset " ,  gsog:Mineral_Material "] ,
                 " mindat:id ":" 52913 ",
                 " identifier ":" 1:1:52913:0 ",
                 " mindat:name ":" Uakitite ",
                 " mindat:ima_formula ":" VN "
         }
         ]
 }
```

Code and results shared on GitHub: https://github.com/quexiang/OpenMindat/blob/main/notebook/Output_DF2File.ipynb



## 5 Discussion

We have fully implemented the designed architecture of the OpenMindat R package and built examples for almost all the developed functions. The R package and its source code were shared on GitHub (https://github.com/quexiang/OpenMindat), together with detailed tutorials on how to install and run the package in the R environment 355 (https://quexiang.github.io/OpenMindat). The first version of this R package (version 1.0.0) was released in the comprehensive R archive network (CRAN) (Hornik, 2012) (https://cran.r-project.org/web/packages/OpenMindat) on February 15, 2024. A list of Jupyter Notebook files (https://github.com/quexiang/OpenMindat/tree/main/notebook), including those shown in the previous section, was also shared to demonstrate the functions and parameters for data query and access from the Mindat API.

What we presented above was from the perspective of the software developers to illustrate the architectural design, the data availability, and the functionality of the OpenMindat R package. Scientists can use those functions flexibly to conduct scientifically meaningful data queries and access tasks. A big advantage of using the OpenMindat R package is that it reduces the scientists' efforts on coding, i.e., with relatively minor coding, they can retrieve a specific piece of data from the Mindat API. For example, using the package, the four use cases discussed by Ma et al. (2023) can each be realized with just a few 365 lines of R code.

1. Retrieve a full list of all IMA-approved mineral species with detailed properties:

```
R> df_ima_minerals <- minerals_ima_list()
```

2. Retrieve a list of mineral species matching certain chemical criteria, such as 'mineral species containing nickel or cobalt, with sulphur but without oxygen':

```
R> df_Ni_S_without_O<- geomaterials_contain_all_but_not_elems(c("Ni","S"), c('O'))
R> df_Co_S_without_O<- geomaterials_contain_all_but_not_elems(c("Ni","S"), c('O'))
R> df_Ni_or_Co_and_S_without_O<- unique (rbind (df_Ni_S_without_O, df_Co_S_without_O))
```

3. Validate alternative mineral/rock names. For example, if the name 'amethyst' is sent, then it would return that the correct mineral species is 'quartz', and that 'amethyst' is a varietal name:

```
R> df_gm_amethyst <- geomaterials_name("Amethyst")
R> df_ima_mineral_name <- mindat_geomaterial_list(ids = c(df_gm_amethyst $varietyof), entrytype=0, ima_status = "APPROVED")
```





4. Provide a hierarchical taxonomy of petrological names and their definitions (e.g., get the rock hierarchy information):


*R> df_gm_rock_parent <- mindat_geomaterial_list(ids = c(''), entrytype=7, fields = c("name","description_short", "rock_parent","rock_parent2"))*

Once the dataset is retrieved, many other packages and functions in the R environment can be leveraged in data visualization
and analysis. Visualizing georeferenced records, such as localities in a map window, is straightforward. **Figure 2** shows a few examples.

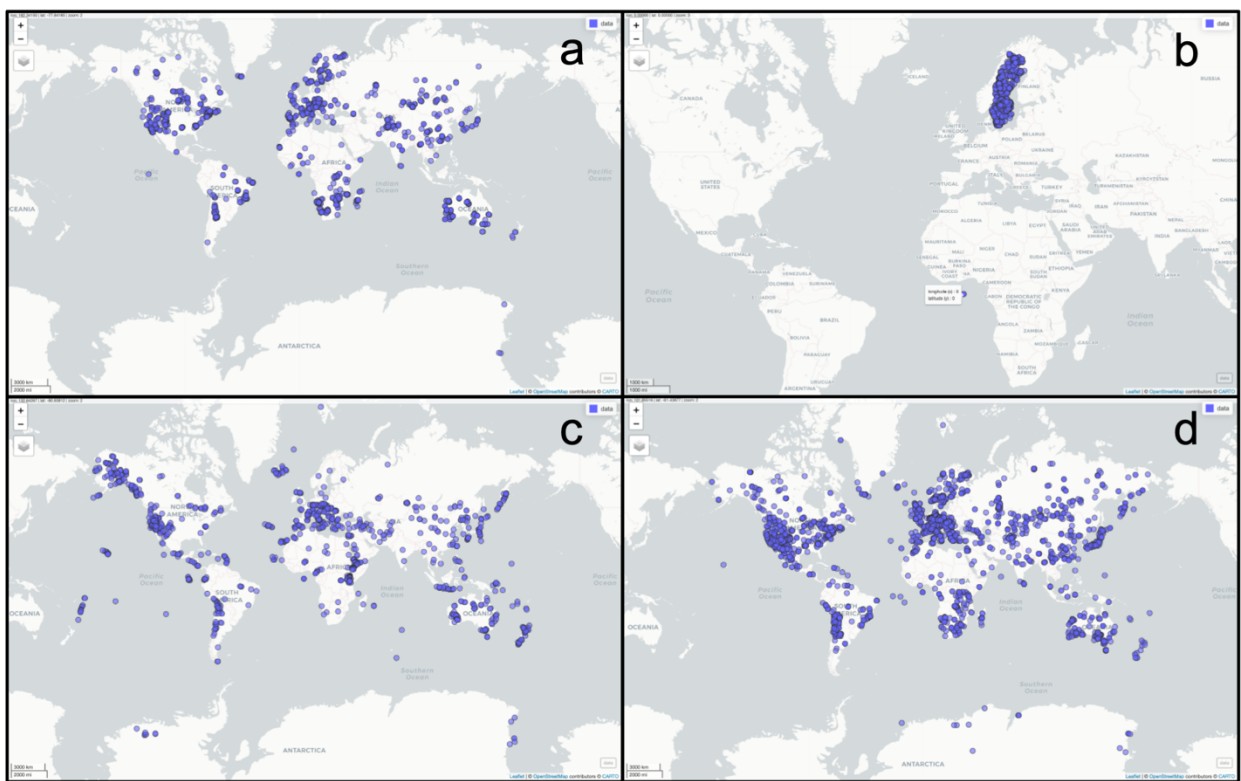

**Figure 2: Mapping locality records retrieved by the OpenMindat R package: (a) As-containing minerals, (b) localities in Sweden,**
**(c) locality descriptions containing 'volcano', and (d) type localities of IMA-approved minerals. Base map © OpenStreetMap contributors 2024. Distributed under the Open Data Commons Open Database License (ODbL) v1.0.**

The Mindat API is constantly upgrading to release more data subjects and fields. Accordingly, we will revise the classes and
functions in the OpenMindat R package. We also collect users' feedback on the Mindat API and the package (from the Mindat
online forum, Slack channels, and direct emails) and incorporate them into our development plans. For example, when this paper was under preparation in November and December of 2023, the Mindat API did not provide access to the mineral



occurrence records (due to heavy data records and concerns about workloads added to the data server). That puts limitations on certain data query conditions. For example, it could not retrieve "minerals that contain cobalt but not oxygen and are found in South Africa or Zambia". The Mindat API technical team (led by Jolyon Ralph) was aware of the users' needs on mineral
occurrence records and has been working on an extension to the API to open the data correctly. Once that extension is fully implemented on the API, we will also make extensions to the R package, such as functions to query locality records of certain mineral species or vice versa.

We can envision a broad variety of applications based on the Mindat API and the OpenMindat R package, such as those in
mineral evolution (Hazen et al., 2008; Hazen et al., 2014), mineral ecology (Hazen et al., 2015), and the co-evolution of geosphere and biosphere (Hazen et al., 2014; Hazen and Morrison, 2020). According to the discussion on mineral informatics (Prabhu et al., 2023), the work plan of the OpenMindat project (Ma et al., 2024), and the vision of the Deep-time Data-Driven Discovery (4D) Initiative (4D Initiative, 2019), a cyberinfrastructure ecosystem based on many open geoscience data resources (including Mindat) will gradually be built to facilitate data-driven discoveries. This includes the databases, software packages,
use cases, and many training activities. Some preliminary work has been implemented. For example, based on the Mindat API and the OpenMindat R package, we recently built an R Shiny app that uses adjacency matrices to explore a variety of correlations in mineral properties, occurrences, and associations (Que et al., 2024). It is foreseeable that, as part of this cyberinfrastructure ecosystem, the machine-readable Mindat data, including this R package and the API, will play an increasingly active role in data-intensive studies.

**6 Conclusions**

This paper presents an R package called OpenMindat for simple, fast, and efficient data retrieval from Mindat, one of the world's largest databases for mineral species and their distribution. This R package is of potentially significant use by various scientists because it bridges the data highway, connecting users' data requirements to the Mindat API server. The machine interface to the Mindat open data enabled by the package will accelerate data-driven geoscience discoveries, as many
geoscientists use the R environment intensively in their work.

This work fills a gap in leveraging technology to expand its underlying cyberinfrastructure ecosystem. The current R package meets most data retrieval needs of Mindat, including retrieval of geomaterials according to chemical properties, physical properties, crystal structures, and more. It also supports data retrieval of localities and IMA materials, and the built-in
compound retrieval functions can support a wide range of application requirements. Moreover, it enables bulk data retrieval and output in various formats, including CSV, JSON-LD, TXT, and TTL, which are popular amongst geoscientists.

Open and FAIR mineralogy data, in terms of mineral informatics, will bring many advantages that revolutionize how we study and understand the Earth (Hazen, 2014; Hazen et al., 2019). Looking forward to the future, as the Mindat API and the OpenMindat R package gradually improve, we hope the Mindat open data will accelerate research and innovation in many research fields, enabling the development of new predictive models, analytical tools, and exploration strategies and leading to many new scientific discoveries.

**Code and Data Availability**

The installation guideline, demos, and documentation of the OpenMindat R package v1.0.0 are accessible at https://cran.r-project.org/web/packages/OpenMindat (Que and Ma, 2024). The code for the OpenMindat R package v1.0.0 can be accessed at the Harvard Dataverse through its DOI: https://doi.org/10.7910/DVN/9NWCDK. The documentation of the Mindat open data API is available at https://api.mindat.org/schema/redoc/. The tutorial on obtaining and using the API token is accessible at: https://www.mindat.org/a/how_to_get_my_mindat_api_key.

**Acknowledgments**

This work presented in the paper was supported by the National Science Foundation, United States (No. 2126315). The authors thank many fruitful discussions within the communities of the Deep-time Data-Driven Discovery (4D) Initiative and the Deep-time Digital Earth (DDE) Big Science Program of the International Union of Geological Sciences.

**Author Contributions**

**Xiang Que**: Conceptualization, Methodology, Software, Writing - Original Draft, Writing - Review & Editing; **Jiyin Zhang**: Methodology, Validation, Writing - Review & Editing; **Weilin Chen**: Validation, Writing - Review & Editing; **Jolyon Ralph**: Data Curation, Writing - Review & Editing; **Xiaogang Ma**: Conceptualization, Methodology, Funding acquisition, Validation, Writing - Review & Editing.

**Competing Interests**

The authors declare no competing interests.

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
