# Peer review of "OpenMindat v1.0.0 R package: A machine interface to Mindat open data to facilitate data-intensive geoscience discoveries"

_EGUsphere, 2024_

## Author Response (AR1)

**Dominik Hezel, 21 Dec 2024:**

The paper presents an R package to query the mindat.org API. Such tools are important and required for an easy, quick and simple access to existing databases. It is difficult to make such tools visible in domain specific journals, as these are not classical research papers, maybe reports, and there might be a categorical gap for such relevant papers. This might be the reason for a couple of shortcomings of the paper, as I outline below. As I will argue, the authors need to make a decision whether they want their work to see published in e.g., JOSS or here.

I find this an important, relevant, and valuable contribution, nonetheless, it needs significant reworking before it can be published.

**Reply**: Thank you for your valuable comments. We hope the manuscript will be published in GMD rather than JOSS. This manuscript presented a data retrieval tool (i.e., OpenMindat R package) based on the Mindat open data API. A primary aim of the tool is to benefit researchers in mineralogy, petrology, and related geoscience disciplines, especially those with limited coding experience. Although JOSS has published many papers about R packages and software, it does not specifically serve the geoscience community. Many JOSS readers may simply overlook this paper as it is domain-specific, and very limited geoscientists read papers published in JOSS. In contrast, GMD has a much higher visibility among geoscientists than JOSS. Mindat is one of the largest databases in geoscience. We hope the OpenMindat tool can lower the barrier of this important data resource to a wide range of potential users amongst the GMD readers. As experts in geoscience, their needs for open data are specific, and their comments and feedback could help improve future versions of the OpenMindat tool.

Following your detailed comments below, we changed the structure of the manuscript and made edits to improve its readability for geoscientists. Specifically, we removed unnecessary technical details about functions in the R package and gave more information about examples that can be realized by the package as well as its features and limitations.

**General comments**

The authors are unnecessarily overselling the capabilities of ML and big data by frequently using superlatives associated with largely unsupported claims. Here are 5 examples just from the beginning:

›... increasingly exciting patterns are discovered in complex earth science big data.‹

›... contains vast amounts of knowledge that are yet to be mined ...‹

›... overwhelming data needs ...‹

›... this can unlock novel insights, enabling groundbreaking discoveries ...‹

›... Mindat plays an increasingly significant role in scientific value and societal impacts ...‹ (how so?)

While I enjoy and support the authors enthusiasm, after a number of years in this field as well, it seems obvious that ML, big data, etc. are important new techniques, but it is not that these new techniques ushered us in a new era of nonstop scientific breakthroughs. I believe tools such as the authors created are nonetheless of great importance and a big service to the community, which, and here I would agree, is slow in picking up such tools that make a geoscientists life much easier. If I would speculate, I would say that the authors fear their contribution might be seen as insufficiently important for a publication if they are not including some bold statements about why it is important. This, I think, reflects a reluctance in the community to embrace new tools such as the ones in this manuscript and appreciate the work that goes into the development of such tools, and their high value and service to the community. I strongly support the publication of such tools, either in specific journals such as JOSS, or, if it is something highly specific to a particular community, in domain specific journals such as this. It might be an opportunity to add a dedicated journal category, such as software/software tool. This would make it much more easier for authors and readers to have such valuable ms.

Hence, I would like to strongly motivate the authors to tune down on the possible, but yet unproven greatness of ML and big data, and focus on the indisputable importance and helpfulness of obtaining more information more quickly and progressing of the underrated importance of making data FAIR. I believe this will make your ms sound more scientific, and less rambling and almost philosophical.

**Reply**: Thank you for your valuable comments and recognition of the value of this study. We have revised the sentences and text in the manuscript to use more objective language to describe our work and its potential value. We do have some concerns that this work may be considered insufficiently novel enough in scientific terms because many domain experts may realize the importance of data to their scientific research but ignore the importance of using corresponding data tools such as this implemented package. Some scientists may overlook its role in facilitating knowledge discovery.

Scientific discoveries or breakthroughs depend on many things. Still, most people believe that at least two aspects of effort are essential: the efforts of domain scientists and the updating and application of new technologies and tools. However, sometimes we overlook the coordination and cooperation of the two aspects: domain scientists need to

understand the updating and application of tools better, and the updating and application of tools need to be more in line with the needs of domain scientists. This inconsistency between the two aspects may become an obstacle to the discovery of specific domain knowledge, and eliminating such inconsistency is often a challenging task, which may be why, although we acknowledge that more new accessible technologies and tools can accelerate the discovery of domain knowledge, we do not know when or to what extent these benefits will arrive. However, a deeper understanding and collaboration between domain scientists and technologies or tools will help accelerate this process. This is also why we want to expose this data tool to the community, aiming to lower the barriers between data providers and users.

I find the current ms not well organised, and although I like the idea of section 4, it is currently not very helpful. But I want to start with an even more general comment. I think the authors need to decide what audience to address: a more technical audience for which they want to provide technical information? If so, this ms should be published in JOSS. If the authors want to address geoscientists who might not yet be (too) familiar with coding, then it can be published here, but the technical terms need to be moved to on single method section, be on a very broad level, and the actual technicalities moved to the online documentation. The authors then need to make it clear in the ms that some kind of access key is required, for which detailed instructions are provided in the documentation (not many will know what a ›token‹ is). Terms such as API, httr, jsonlite, etc. all need to be moved into the documentation. The ms then needs less technical details (unless the authors choose JOSS), and present what the tools does on a much more practical level. In case the authors want to publish their tool here, I suggest the following:

Section 1 needs some statements why using such a package is advantageous compared to using the mindat webpage. And here I mean concrete, practical statements, not the claims mentioned above. This will be helpful for researchers unfamiliar with such possibilities.

**Reply**: Thank you very much for your valuable comment. We moved the detailed technical content to online documentation and reorganized the manuscript to make it more clear and friendly to geoscientists. We have revised Section 1 and added 3 advantages of using our package (vs. the Mindat webpage): First, although Mindat has established human-readable web portals that provide large numbers of pages for displaying information on mineral species, localities, etc., they are not machine-accessible.  The software package we developed provides rich machine-readable interfaces, which can better serve large-scale data access of analytical machines. Second, although Mindat provides a webpage portal (https://www.mindat.org/advanced_search.php ) to filter data by specifying

constraints interactively, filtering data with multiple conditional constraints requires multiple interactions. Moreover, the interactive data filtering portal can only support browsing rather than downloading. However, the analysis of many scientific issues requires the use of filtered data, not just browsing. Third, sometimes the webpage cannot load all the filtered data records at once, or it may not display them in the form we want (such as sorting). Using the package tool can easily display data according to our needs. Please see lines 44-67.

Fig. 1 is a good idea, but needs to be reduced to a much simpler version that can be understood in an instant. The more complex figure can be part of a documentation.

**Reply**: Thank you very much for your valuable comment. We replaced the previous complex Figure 1 with a new simple one, which makes this manuscript clearer and more concise to geoscientists. The previous complex figure has been moved to the online document for those who are more inclined to pursue technical details.

Section 3 is a mishmash of technicalities that need to be moved into section 2, while the capabilities can easily be merged with a new section 4.

**Reply**: Thank you for your comment. We reorganized Section 2, removed the previous Section 3, and merged parts of it and the previous Section 4 into a new Section 3 in the revised manuscript.

Section 4 should be completely rewritten. Examples are great, but should be presented so that even non-programmers understand it and might be motivated to use programming tools in the future. First, there are way too many examples. As a programmer, you will get the idea instantly and start using the tool, as a non-programmer you get lost. I suggest focussing on what the tool does on an abstract level, which is, as I understand it: getting information from mindat using predefined functions, i.e. using a filter to only get specific data. Here a single line of such a function (cf. line 256), i.e., filter, could be used, and explained what the various elements of the function does. A programmer will not read the explanation, but understand what is done from this one line, and a non-programmer will fully understand what is done. Then I suggest to prepare an example e.g., Jupyter Notebook with this function and the result of this function. Provide a screenshot so anyone get a real idea what is happening. Then reduce Table 1 to fewer functions, e.g. 5-10, and explain all of these in plain words and what they do. I would remove all other tables of section 4. Then

explain the additional functionalities, such as combining functions/filters, again with one line of example code and a screenshot of this line and its result from the example Jupyter Notebook. Do the same with the ›wild cards‹ capabilities. This way you have a very clean, straight forward, much briefer and clearer section 4. Also a Jupyter Notebook with examples provide a great start for a user to actually try your tool. I would find it fully sufficient if the explanation how to get a mindat token would be at the beginning of such a notebook. Using tools such as this are much more instructive than these rather convoluted tables. If someone is really interested, I bet she or he would rather try it than read through all these pages. The few screenshots should sufficiently convey the idea and potential. Just add a brief description how to get started with the Jupyter Notebook. That might in fact be crucial.

**Reply**: Thank you for your valuable comment. We completely rewrote Section 4 following your suggestion. We removed technical implementation details and presented this section with limited examples to make them easier to understand for non-programmers. Table 1 was simplified, and the functions are described in plain words. We also removed most of the previous examples and tables and described the functions of encapsulated classes in summary form. We kept a few examples to demonstrate how to call functions to get specific data to meet users' needs. Corresponding codes and results are also presented in Jupyter Notebooks, and a brief description was also included.  For introductions on how to setup up Jupyter Notebook  for the R environment, we added the following link for reference (listed in Table 10): https://developers.lseg.com/en/article-catalog/article/setup-jupyter-notebook-r. Users can also find instructions on how to get started with this package on our GitHub.

This tool can help geoscientists convert data requirements into corresponding http requests and send them to the Mindat server for response. However, the prerequisite is that geoscientists need to obtain permission from Mindat first. The Mindat team uses a security mechanism (aka token) to avoid malicious access (or hacker attacks), thus keeping their data safe. Therefore, it should be noted that any user who needs to use this package, should first apply for a token from the Mindat official website (https://www.mindat.org/ ). However, it is easy to get this token, which can be obtained through the link: https://www.mindat.org/a/how_to_get_my_mindat_api_key.

Some ideas for more specific examples: at the moment, there are only these tables, which are rather convoluted. Why not providing some simple e.g., pie charts that show some statistical results such as the relative numbers of mineral crystallographic types, relative

numbers of minerals from each country, or the like. Or similar bar charts. You already have a map that is also a much more interesting result than the tables. All this would make the examples much more lively and interesting.

**Reply**: Thank you for your valuable comment. We have rewritten the sections on examples and results to present our retrieved data records in a summary format. We also added some examples of visualization of the retrieved records to make them easier and more interesting for readers to understand. Please see Section 3 of our revised manuscript.

I couldn't find a demo on ›https://cran.r-project.org/web/packages/OpenMindat/‹ as promised in line 439, at least not quickly and easily. I did find code examples on https://quexiang.github.io/OpenMindat/, but only after trying several links (there are quite a couple, maybe you could collect and describe them in one, single table, at the moment it is a bit left to the reader to find his or her way through the various links, demos, ... or provide just one link to one page from which you link out to the others) could not readily run them. Maybe give some advice to python coders like me. In general, this page is great, and could contain the documentation as suggested above.

**Reply**: Thank you for your valuable comment. We've created a list of resources and links related to this package based on your suggestion. This makes it more friendly to readers and users.

Finally, I really would be interested about speed and limitations. This is only mentioned peripherally in line 295. But this would really be some critical information: How long does it take to query properties of all the minerals, can I easily display the images of 500 minerals, what cannot be done, etc. would be valuable information. The authors claim in the title that with their package ›*data-intensive geoscience discoveries*‹ are possible. So I would be interested to what extent data-intensive queries are possible. E.g., also, whether I can do 10, 100, or 1000 such queries a day. Maybe even in class with 20 students.

**Reply**: Thank you for your valuable comment. The Mindat open data API is maintained by the Mindat technical team. They review and permit user registration requests, monitor the status of the server, and defend against cyber-attacks or malicious mass downloads. For individual researchers, the default API usage limit is 1,000 requests per hour. Based on our experience in the past two years, that should be enough to meet the needs of most people. Specific users who need more frequent and larger data access can contact the Mindat technical team for permission. Despite its benefits and potential, the OpenMindat R package faces certain limitations and needs further extension. For instance, the current

version does not support queries involving mineral occurrences and photos due to restrictions in the Mindat API. This limitation constrains studies that require detailed spatial analysis of mineral distributions. For example, retrieving "minerals that contain cobalt but not oxygen and are found in South Africa or Zambia" is almost impossible or may require complex commands for the current version of package. Additionally, some users may encounter challenges in navigating the package's advanced features, underscoring the need for more detailed tutorials, examples, and user support. To address these issues, the development team is actively working on cleansing data and expanding the package's functionality. Planned updates include incorporating mineral occurrence records as the API evolves, enhancing the package's documentation, and developing interactive tutorials to guide users through complex queries. We are also collecting feedback from the geoscientific community to shape these improvements. (Please see lines 290-300)

Section 5 in fact resembles to what I am suggesting above. I apparently wasn't aware of this while suggesting the above, which might happen to another reader as well, hence a restructuring might be sensible. If an example from a co-author is used (Ma et al. 2024, I guess, not 2023 as in line 364), this might be made more clear. Also, are this valid code lines? I am familiar with python, not R. It looks like this are mixtures of results and commands, but I might be mistaken. Should I be right, I would suggest to only use codelines, I find this combination confusing. Otherwise section 5 is more of a summary than a discussion with another example.

**Reply**: Thank you for your comment. Following your suggestions, we have rewritten sections 2,3,4 and 5. Some previous content of these sections was reorganized to make the conclusions and discussion clearer. Those lines that begin with "R>" are executable codes in the R environment. Some of the previous example code lines and tables have been adjusted to make them more understandable to non-programming readers.

Section 6 concludes with some additional, exhaustive announcements what such tools might be able to do, which is no less than ›... bring many advantages that revolutionize how we study and understand the Earth ...‹. Again, I am all for it, but – see above.

**Reply**: Thank you for your comment. We have rewritten Section 6 (Section 5 in the revised manuscript) to summarize our work more clearly. (See lines 301-314)

**Detailed comments**

1. Title: I am not sure I understand what is meant by ›machine interface‹. I would likely have expected simply: The OpenMindat R package to facilitate data-intensive mindat.org queries«

   **Reply**: Thank you for your valuable comment. Machine interface means it allows the other applications to interact with the server or system.  Although Mindat web portals can support human-readable and provide large numbers of pages for displaying information on mineral species, localities, etc., it still faces several challenges, especially on data access. We listed 3 advantages of using the machine-readable OpenMindat R package (vs. the Mindat webpage). Please see our reply to the third comment.
   Through the machine interface provided by this package, users can do a large amount of data from the Mindat API server, making it easier and more confident to conduct data-intensive analyses and applications. For example, an R shiny application that we developed previously for mineral information exploration based on adjacency matrices ( https://quexiang.shinyapps.io/Adjacency_Matrix_4_Mineral_Informatics) currently still using local database data and offline data, but in the future, such application can use this package to read data directly from the API server. Thus, users or analysts don't have to worry about the results of these applications being based on outdated data, as this data is directly accessed through the Mindat API.

2. 41f and 69ff: I suggest to make a table on what is stored in mindat.

   **Reply**: Thank you for your comment. We have listed the contents stored in Mindat in a table (Table 1 in our revised ms). However, since the Mindat API currently only supports some open data that has been cleaned by the team, data such as photos and occurrences are not yet available. We will also coordinate with the Mindat team to gradually expand the data retrieval function of OpenMindat in future work.

3. 61ff: I suggest to delete this table of contents. It is not needed and likely therefore uncommon.

   **Reply**: Thank you for your comment. We have removed the last paragraph of section 1 based on your suggestion.

4. 91: I think this IMA description is not needed.

**Reply:** Thank you for your comment. We deleted the IMA description and revised the sentence to make it more concise. Please see lines 92-94.

5. 93: IMA, not IAM.

   **Reply:** Thank you for pointing it out. Changed as suggested (line 94).

6. 103: The abstract says JSON-LD, here it says TXT.

   **Reply:** Thank you for your comment. This package provides functions to save the retrieved data in different formats, including JSON-LD and TXT. We changed the description to make it consistent.

7. 123: I am not sure what is meant here in (2).

   **Reply:** Thank you for your comment. The Mindat Cache class is an inner class, that is designed to avoid repeat data queries from the API server and improve query efficiency. This cache manages the generating, storing of cached data, and destoning of different cache instances. For example, the token of the Mindat API will be cached after the Mindat connection is first established, so when executing query functions, it does not require inputting the token again. We hide some technical content and focus more on the external classes, as you suggested in previous comments. We move some technical details to our online document.

8. 133: cf comment to line 103.

   **Reply:** Thank you for your comment. Same as the comments on line 103, which have been processed consistently.

9. 149: what are hitter and jsonlite, how are these installed?

   **Reply:** Thank you for your comment. "httr" and "jsonlite" are two R packages required to run OpenMindat. They are mainly used to send http requests and to parse json format data. Similar to other R package installations, you can call the install.packages() function. For example, to install httr, we can call install.packages("httr") and then load it through library(httr). If we use a remote package installation (not downloaded and installed locally), the dependent

packages are already installed by default, and we only need to call "library" to load them. We have detailed installation instructions in GitHub, and users can configure the code execution environment according to the guidance.

10. 210ff: I don't understand what the ›special symbols‹ are.

    **Reply:** Thank you for your comment. Some query functions in the package can handle symbol parameters, such as "+", "-", etc. For example, the "geomaterials_opticalsign". If we would like to obtain the geomaterial records that have negative optical signs, we can execute the following code line:

    ```
    geomaterials_opticalsign("-")
    ```

11. 225: What are ›head 3‹ records?

    **Reply:** Thank you for your comment. It means the first 3 records of the retrieved data. This expression is no longer retained in our revised manuscript. We have removed most of the retrieved records from the previous table and replaced them with descriptions of them, which can express our content more concisely. For detailed returned results, users can still refer to our online example Notebooks.

12. 253: ›combine *conditions*‹, I guess ›property‹ would be more appropriate than ›conditions‹

    **Reply:** Thank you for your comment. Changed as suggested.

13. 263: This is not a helpful link at all. It just points to the API description which coders likely already know, and others will not understand.

    **Reply:** Thank you for your comment. We have revised the description here to make it clearer so that users can more accurately refer to the relevant fields and instructions.

14. 289: Is it not possible to include it now?

**Reply:** Thank you for your comment. As far as we know, the updated Mindat API server can provide mineral list information now, but function access also faces some challenges, the current released R package function can only partly support it. We added an example to demonstrate how to obtain the top 10 IMA-approved minerals found in a specified country (e.g., Canada). We expect to provide more functions to support it more friendly in the next version. Please see lines 183-195, and 291-301.

15. 325: yet another list of output-formats

    **Reply:** Thank you for your comment. They are all the export formats that the current package can support. We may expand support for more export formats, such as XML, as needed.

16. 352: What kind of examples? I could not find examples that I can play around with.

    **Reply:** Thank you for your comment. Yes, technically, the code examples in these notebooks can be run directly on online platforms such as Colab, but it is not recommended because we are not authorized by Mindat to provide public tokens. It is recommended to apply for a personal token form the Mindat and then run the functions provided by the R package to get the data they need.

17. 353: ›are shared‹

    **Reply:** Thank you for pointing it out. Changed as suggested.

18. 357: Provide just one webpage and link out from there, all these links are confusing. I suggest using only: https://quexiang.github.io/OpenMindat/index.html

    **Reply:** Thank you for your comment. Changed as suggested.

19. 409: When you say envision, I would expect what could be done in the future, not the works of more or less one first author from what has been done so far.

    **Reply:** Thank you for your valuable comment. We have rewritten the Discussion section to remove descriptions like this, making the manuscript more concise and clearer. Please see lines: 279-290.

20. 431: And another combination of output formats. Maybe this does not need to be mentioned that many times anyway.

**Reply:** Thank you for your comment. We have deleted this redundant description in the manuscript.

**Anonymous Referee #1, 25 Nov 2024:**

General comments

The manuscript describes an R package as the machine interface to the open data of Mindat.org. So, it is not a model itself but describes tooling around models to access data. Overall, I think that there should be more papers like this that describe the details behind software packages, in journals that scientists that are not developers can find, read, and understand. This paper clearly states that it is written from the software developer's perspective. I wasn't sure if this was in scope for Geoscientific Model Development, as sometimes these explanations are left to the grey literature user manual, or a software journal like JOSS, but I found other examples of similar papers in GMD, so I believe it is in scope.

Reply: Thank you for your valuable comment. We revised our manuscript to make it more friendly to non-programming scientists. We look forward to a deeper understanding and collaboration between domain scientists and tool developers, accelerating the data-driven mineral knowledge discovery process. This is also why we chose the GMD journal. GMD has higher visibility among geoscientists than JOSS. We hope that more scientists can find and use this tool to retrieve the data records they need and explore the hidden knowledge therein, thereby promoting the development of earth science.

Coming from a geoscientist, and not software developer background, some of the concepts and terminology were difficult for me to understand. However, I do think that the paper is useful to be published, with some efforts made to increase the understandability that might be more obvious to software developers (where to find the controlled vocabularies, brief descriptions at the top of .ipynb). Also, I am not an R user so I hope there are other reviewers that are able to execute the examples, my review is based on the paper and looking at the available code.

The paper itself references several examples showing uses and potential uses. The research group and its collaborators have built up a robust technical ecosystem for this type of data exploration and scientific research, with leaders in the field. Having these newer methods used by more people who access mineral data would be a strong positive.

The methods are aligned with the best practices that I know about. This is a minor comment but I found the description of the benefit a bit exaggerated in language and benefit. This might be because at my organization we are held back from effusive language and told to just state the facts. However, this is up to the editor, it is a style comment. (Examples: overwhelming, groundbreaking, viral, invaluable.) For the review prompt question, "Is the language fluent and precise?" these would be examples of imprecise language.

**Reply:** Thank you for your valuable comment. We have revised some of the expressions and used more concise and appropriate words to describe them. We have reorganized the structure and content of the article, hidden some detailed technical implementations (moved to online documents or other technical description materials), used a simpler framework (new Figure 1) and expression, and deleted some obscure professional terms. We have also adjusted some previous examples and tables, hidden most of the previous data retrieval results, and explained more about the execution results so that geoscientists can more easily understand and apply the tools we developed.

Regarding reproducibility, the repository appears to have everything needed to reproduce, with quite decent documentation, but I hope there is another reviewer that can confirm that.

The tables, in the format presented in the preprint, are not in the best format for readability. For this type of content I would hope that the final version they are fixed-width.

**Reply:** Thank you for your valuable comment. These sample codes are executable, and users can refer to our examples or the operation manual provided by the R package to implement the data retrieval tasks they need. We adjusted most of the original tables to make them simpler, which also facilitated the final layout.

All other components of the paper (abstract, background, references, supplementary material) appear to be adequate and useful to me.

Seems like this type of capability could be generalized to many scientific database APIs, so it is important that the paper is understandable to non-mineralogists.

I like how there are a variety of output formats, and that it looks like a lot of user-research was done up to this point. This gives me more confidence that this product is usable and relevant to the community.

 **Reply:** Thank you for your valuable comment. This model of database API and software package may be an effective model for community open data and can be used as a reference for open data in other disciplines. Our R package currently supports data file output in some common formats. Subsequent package updates can expand support for more formats (depending on needs), such as XML, etc.

Technical corrections

1. The language is a little bit more opinionated/flowering than allowed in my institution "groundbreaking" etc. - leave it to the editors to decide. Simpler more direct language would be best for the international audience.

   1. overwhelming, viral,

   2. data highway

   **Reply:** Thank you for your valuable comment. We revised the sentences throughout the article to avoid inappropriate words and make the article more suitable for international audiences.

2. Line 51 - cite reference for FAIR principles

   **Reply:** Thank you for your valuable comment. Revised as suggested. Please see line 53.

3. Line 63 - is this first use of IMA? Please expand the acronym, I see it is expanded later, but expand acronyms at first use.

   **Reply:** Thank you for your valuable comment. We revised the manuscript and used IMA's full name for the first time it appeared. Please see lines 31-32.

4. Line 71 - what are meteoritiecs and petroleum categories and how are they related?

**Reply:** Thank you for pointing out the incorrect statement. It has been corrected. Please see Table 1.

5. Are all of the Mindat fields, vocabularies, etc. defined elsewhere, make sure to reference it clearly for new users.

   **Reply:** Thank you for your valuable comment. We have a clearer explanation in our revised manuscript. We also summarized Table 10 to help readers quickly find relevant documentation and instructions, including the descriptions on the fields of geomaterials: https://github.com/smrgeoinfo/How-to-Use-Mindat-API/blob/main/geomaterialfields.csv . Since the Mindat API is still in the data cleaning and development stage, its corresponding online documentation may not be detailed enough. We are also working with the Mindat team to improve it so that users can better understand it. Please see lines 296-301 and Table 10.

6. Figure 1 is informative but would be even more informative if it followed a pathway starting from the user request. (rearrange it so that there is a path for the reader to follow)

   **Reply:** Thank you for your valuable comment. We have updated Figure 1 to make it look simpler and attached an example to help readers better understand how it works. We have moved the previous more complex technical diagram to the backend (GitHub documentation) for more technical users. Detailed configurations and how to get started are also described on our GitHub.

7. Line 110 - define what "classes" and "functions" mean in this context, as users are probably a very broad range, some who may not be familiar.

   **Reply:** Thank you for your valuable comment. Thank you for your valuable comment. The implementation of the R package is based on object-oriented programming, where a class can be seen as an abstraction of a subject (some classes were designed to correspond to the different data themes provided by the Mindat API), and a function is a class method that specifically performs to satisfy a certain data theme. We have made corresponding additional explanations in the revised manuscript, please see lines:114-131.

8. Table 1: something about the spacing is a little confusing of how the different columns line up.

   **Reply:** Thank you for your comment. We reduced the line spacing in all tables and added some horizontal lines to separate different classes, making the manuscript easier to read.

9. Table 1: The caption says "some of the functions" and the text says 100 functions. Where does one go to view ALL of the functions? Is there a repository/Github link that is comprehensive?

   **Reply: Thank** you for your valuable comment. There are serval ways to view all of the functions: First, call function "help (package = OpenMindat)" in the R environment (R Studio). Second, view the reference of our tutorials: https://quexiang.github.io/OpenMindat/reference/index.html . Third, view the reference manual of the CRAN R package: https://cran.r-project.org/web/packages/OpenMindat/OpenMindat.pdf.   We have added some corresponding instructions to our revised our revised manuscript. Please see lines:100-101 and Table 10.

10. Lines 120- 140, split up into paragraphs or bullets for clarity.

    **Reply: Reply:** Thank you for your comment. We rewrote the paragraph to make it concise. Please see lines 100-111.

11. Line 178: use of "besides" is awkward- another word would be more appropriate. Maybe "Also"?

    **Reply:** Thank you for pointing it out. We rewrote this section and removed the description in our revised manuscript.

12. Table 2: "Head 3 records" does that mean the first three records? Use more straightforward language for this venue with these readers. Everywhere in the paper where it says "head 3" can it be replaced by "first 3"?

    **Reply:** Thank you for your valuable comment. Yes. It means the first 3 records. To make it easier for non-programming readers, we have readjusted the tables in the manuscript, removed this description of data records, and replaced it with a

paragraph of descriptions. Technical readers can view the retrieved data records through the online notebook, whose URL was attached to the updated table.

13. Table 2: is there a typo in the first row? There are special characters and unicode decimal codes.

   **Reply:** Thank you for your valuable comment. This may be an issue with the data storage in the Mindat API. The query function in our R package only displays all retrieved data in text form. We will give feedback to the Mindat team and check if these special characters need to be converted or handled when outputting.

14. There are R examples and there is a ipynb example - could it be more clear which or both? I think that the ipynb example is R, but can that be clarified in a short header section of the .ipynb?

   **Reply:** Thank you for your valuable comment. "*. ipynb" is the file format of Jupyter Notebook, which can support both R and Python (we use R in our examples). Because the file can save the executed code lines, comments, and results, it is friendly for demonstration and reading. However, we have used some R code lines (starting with ">R") in the manuscript as examples, and these are simple R commands (i.e., function and its arguments (if any)) without comments and results. We have rearranged the example functions and their corresponding notebook links in a new table and hidden the results to make them clearer.

15. Line 360 is clear that it is from software developer perspective, that is useful.

   **Reply:** Thank you for your valuable comment. We have revised our manuscript and corresponding content to make it more readable by non-programmers.

16. It would have been useful if more fixed-width font were used in the PDF. Does this happen at the journal editorial stage?

   **Reply:** Thank you for your comment. We have revised the manuscript and adjusted some typography to make it look more consistent. Some adjustments and proofreading should be required before formal publication.

17. I had some trouble repeating the resolution of the .ipynbs in the tutorials repo, multiple times the notebook did not resolve, but other times it did.

    1. For example: https://github.com/quexiang/OpenMindat/blob/main/notebook/Retrieve_Geomaterials_by_physical_prop_1.ipynb

        1. The first time I get the "unable to render code block"

        2. upon reloading a second (or third) time, it worked. Is this a known issue or repeated by anyone else? It happened on multiple machines, in the Chrome browser, for me.

    **Reply:** Thank you for your comment. Similar reports of rending failures have also occurred in some other repositories on GitHub. This is not the issue of the "*. Ipynb" file. I think this should be the loading failure caused by the slow network of the repo on GitHub. This file is relatively large at 3.65MB and rending it directly in the browser through GitHub may require a good network connection and take a long time. Users can download it locally and then open it in jupyter notebook.

18. For the examples in the notebooks on Github, can you place a header at the top of the .ipynb that describes that these are examples for OpenMindat in R? Then a sentence or two of what the notebook does, what the output says, and where to get more information if you wanted to be able to understand the output better. (As someone who is not very familiar with OpenMindat, I would want to know where to access information about interpreting the returned results.) Especially since there are examples for python too, I think it would be useful for each .ipynb to have a short description at the top.

    **Reply: Thank** you for your valuable comment. Following your suggestion, we have included a short description for each *.ipynb, which is easier for users to use and read. Please see those files shared on our GitHub: https://github.com/quexiang/OpenMindat/tree/main/notebook .

---

## Referee Report (RR1)

Review of »OpenMindat v1.0.0 R package: A machine interface to Mindat open data to facilitate data-intensive geoscience discoveries«

by Que+

**reviewer: Dominik Hezel**
**ms received: 05.02.2025**
**review completed: 04.03.2025**

The authors greatly improved their manuscript compared to its first version, and I find this an exciting, highly important, and interesting read – but even more so, a fantastic R package. I only have some minor comments below, and am now looking forward to finally see this paper published.

**Detailed comments**

1. 59: I was excited to read there might also be a python package, but this is only mentioned here. So either delete or make some more statement here or later that there is also a python package in the works, and provide some hint to when this might be released.

2. Table 1: Why is the meteoritics volume number in red?

3. 84: I find the term ›address‹ confusing. Could this be simply replaced by e.g., ›hierarchy‹? A hierarchy can have levels, so this might be more descriptive than ›address‹.

4. 87: ›tectonic plate‹?

5. 88ff: The bracket starting with ›(Studies of …)‹ is confusing. Maybe this could be made as a proper sentence that has a better connection to its previous sentence.

6. 114: Delete ›above‹

7. 115: Delete ›us‹

8. 119: Delete extra space before ›it supports‹

9. 173: ›This function category …‹? cf. your Table 6

10. 184: ›… few lines of code …‹ would be more idiomatic

11. 224: ›… code are shared …‹

12. 231: ›… section, is also shared …‹

13. 233: Delete ›those‹

14. 235: I don't think the tool is reproducible, but it allows for reproducible data query. Rephrase accordingly.

15. 239: ›interactions‹ instead of ›interventions‹

16. 239 ›webpages‹ or rather ›webpage‹?

17. 242: ›… programming skills, but require large datasets …‹

18. 250: The FAIR principles and open science are two different things. An e.g., database can be FAIR but not open, i.e., there is e.g., a paywall. This is not overly important in this context, but regarding these principles and concepts are still new to many, I would suggest to make it clear that these are different things.

19. 251: ›… is increasingly demanded in the …‹

---

## Author Response (AR2)

**#Reviewer1 (Report #2)**

This is my second review. The authors did a very good job of addressing both reviewers' comments. I found it much easier to understand and follow and think it will be the same for many readers.

I only have very minor wording suggestions for the authors' consideration.

L 11: "had been" -> was

Reply: Revised as suggested. Please see Line 11.

L45: constraints: parameters

Reply: Revised as suggested. Please see Line 46.

L63: Would the platform be R Studio, and RMarkdown the framework? Just suggesting something more in line with/parallel to Jupyter. Not sure

Reply: Thank you for your comment. We replaced the RMarkdown with R Studio. Please see Line 63.

Table 1: use of "volume" is a little awkward. Since there is no unit, I would prefer "number," but this is just a preference.

Reply: Thank you for your comment. The title was changed as suggestion. Please see Lines 72-75.

Table 1: some definitions are not meaningful definitions to me ("occurrence records" and "literature references")

Reply: Thank you for your comment. We revised the description of occurrences and remove the row of "literature references". Please see Lines 73-75.

L 75: highlighted phrase does not make sense to me. "Currently, it provides a separate access endpoint for each data subject." - I don't know the significance of that. Maybe explain in a sentence or leave out.

Reply: Thank you for your comment. As the number of subjects and records in the crowdsourced Mindat database increases, the API server may open or update its data access endpoints. Some new features of our package will be updated according to the endpoints provided by the server. We have added some descriptions in the revised manuscript. Please see Lines 77-79.

L86: The description of the hierarchy and what is at the "top" or "broadest" is a bit confusing to me. Somehow I thought top of the hierarchy would be most detailed. Not sure if it can be reworded to reduce confusion, but it is okay the way it is.

Reply: Thank you for your comment. The top level of locality (0) is defined by the mindat.org database, and we just follow its principles here to keep the concepts consistent. Please see Lines 88-89.

L87: plate = tectonic plate?

Reply: Thank you for your comment. Yes. We used "tectonic plate" in our revised manuscript. Please see Line 89.

L95: strange to call CSV a specified format for these particular cases since it is the most general format, but I guess I get the point. Might be better to list the different formats for and say they are for different use cases.

Reply: Thank you for your comment. We have made changes based on your suggestions. Please see Lines 97-98.

Figure 1: typo in cotain = contain

Reply: Thank you for pointing this out. We revised it and updated the Figure 1. Please see Line 99.

Figure 1: better, but would be even more improved if each step was labeled 1-4 to show the sequence of events.

Reply: Thank you for your comment. We revised it and updated the Figure 1 as your suggestion. Please see Line 99.

L 103: remove "and it"

Reply: Thank you for your comment. Changed as suggested. Please see Line 106.

L 105: "records"?? better word?

Reply: Thank you for your comment. We use "consists of" instead of "records". Please see Line 108.

Table 3: typo in filename "wildcar" vs. "wildcard"

Reply: Thank you for your comment. We have revised the filename and updated Table 3. Please see Line 141.

L 139: says "three tasks" but there are two examples. Maybe the two examples do three things, but it is confusing, reword to reduce confusion.

Reply: Thank you for pointing this out. We only showed two examples and have revised the relevant statements accordingly. Please see Line 142.

L164: write out the element As after the abbreviation

Reply: Thank you for your comment. Changed as suggested. Please see Line 167.

L173: "the record retrieval" -> "retrieval of records"

Reply: Thank you for your comment. Changed as suggested. Please see Line 176.

L288: potentials -> potential

Reply: Thank you for pointing this out. Changed as suggested. Please see Line 294.

L289: this line seems like it got jumbled and needs to be reworked "the current version does not friendly support queries involving mineral occurrences." maybe just swap the two words friendly and support. or "the current version does not support user-friendly queries..."

Reply: Thank you for your comment. Changed as suggested. Please see Line 295.

**#Reviewer2 (Report #1)**

reviewer: Dominik Hezel
ms received: 05.02.2025
review completed: 04.03.2025

The authors greatly improved their manuscript compared to its first version, and I find this an exciting, highly important, and interesting read – but even more so, a fantastic R package. I only have some minor comments below, and am now looking forward to finally see this paper published.

**Detailed comments**

1. 59: I was excited to read there might also be a python package, but this is only mentioned here. So either delete or make some more statement here or later that there is also a python package in the works, and provide some hint to when this might be released.

Reply: Thank you for your comment. We did develop a Python package to meet the needs of different development users, but considering its implementation mechanism is different; to avoid misunderstanding, we deleted the description here. Please see Lines 59-60.

2. Table 1: Why is the meteoritics volume number in red?

Reply: Thank you for pointing this out. We merged and checked the data and forgot to change it back to black. It has now been changed to black. Please see Line 74.

3. 84: I find the term ›address‹ confusing. Could this be simply replaced by e.g., ›hierarchy‹? A hierarchy can have levels, so this might be more descriptive than ›address‹.

Reply: Thank you for your comment. Following the principle of locality hierarchies in Mindat, we will replace the word "address" with "locality" and explain its hierarchical structure. This should make it clearer for readers. Please see Lines 88-89.

4. 87: ›tectonic plate‹?

Reply: Yes. We changed it to "tectonic plate". Please see Line 89.

5. 88ff: The bracket starting with ›(Studies of …)‹ is confusing. Maybe this could be made as a proper sentence that has a better connection to its previous sentence.

Reply: Thank you for your valuable comment. We removed the brackets and used them as an example to explain the previous statement. Please see Lines 91-93.

6. 114: Delete ›above‹

Reply: Thank you for your comment. Changed as suggested. Please see Line 117.

7. 115: Delete ›us‹

Reply: Thank you for your comment. Changed as suggested. Please see Line 118.

8. 119: Delete extra space before ›it supports‹

Reply: Thank you for your comment. Changed as suggested. Please see Line 121.

9. 173: ›This function category …‹? cf. your Table 6

Reply: Thank you for pointing this out. We have revised the content of the function category in Table 6 to make it more accurate. Please see Line 183 (Table 6).

10. 184: ›… few lines of code …‹ would be more idiomatic

Reply: Thank you for your comment. Changed as suggested. Please see Line 187.

11. 224: ›… code are shared …‹

Reply: Thank you for your comment. Changed as suggested. Please see Line 225.

12. 231: ›… section, is also shared …‹

Reply: Thank you for your comment. Changed as suggested. Please see Line 234.

13. 233: Delete ›those‹

Reply: Thank you for your comment. Changed as suggested. Please see Line 236.

14. 235: I don't think the tool is reproducible, but it allows for reproducible data query. Rephrase accordingly.

Reply: Thank you for your comment. Changed as suggested. Please see Lines 238-239.

15. 239: ›interactions‹ instead of ›interventions‹

Reply: Thank you for pointing this out. Changed as suggested. Please see Line 242.

16. 239 ›webpages‹ or rather ›webpage‹?

Reply: Thank you for your comment. Changed as suggested. Please see Line 242.

17. 242: ›… programming skills, but require large datasets …‹

Reply: Thank you for your comment. Changed as suggested. Please see Line 245.

118. 250: The FAIR principles and open science are two different things. An e.g., database can be
FAIR but not open, i.e., there is e.g., a paywall. This is not overly important in this context,
but regarding these principles and concepts are still new to many, I would suggest to make it
clear that these are different things.

Reply: Thank you for your valuable comment. According to your suggestions, we have added relevant statements to make the manuscript
clearer and more complete. Please see Lines 250-255.

19. 251: ›… is increasingly demanded in the …‹

Reply: Thank you for your comment. Changed as suggested. Please see Line 257.